# *Euterpe oleracea* Mart. Bioactive Molecules: Promising Agents to Modulate the NLRP3 Inflammasome

**DOI:** 10.3390/biology13090729

**Published:** 2024-09-17

**Authors:** Carolina Bordin Davidson, Dana El Soufi El Sabbagh, Amanda Kolinski Machado, Lauren Pappis, Michele Rorato Sagrillo, Sabrina Somacal, Tatiana Emanuelli, Júlia Vaz Schultz, João Augusto Pereira da Rocha, André Flores dos Santos, Solange Binotto Fagan, Ivana Zanella da Silva, Ana Cristina Andreazza, Alencar Kolinski Machado

**Affiliations:** 1Graduate Program in Nanosciences, Franciscan University, Santa Maria 97010-030, RS, Brazil; carolina.davidson@ufn.edu.br (C.B.D.); sagrillomr@ufn.edu.br (M.R.S.); julia.schultz@ufn.edu.br (J.V.S.); andre.santos@ufn.edu.br (A.F.d.S.); sfagan@ufn.edu.br (S.B.F.); ivanazanella@ufn.edu.br (I.Z.d.S.); 2Laboratory of Cell Culture and Bioactive Effects, Franciscan University, Santa Maria 97010-030, RS, Brazil; amanda.kolinski@ufn.edu.br; 3Department of Pharmacology and Toxicology, University of Toronto, Toronto, ON M5G 2C8, Canada; dana.elsoufielsabbagh@mail.utoronto.ca (D.E.S.E.S.); laurenpappis@gmail.com (L.P.); 4Department of Biochemistry and Molecular Biology, Federal University of Santa Maria, Santa Maria 97105-900, RS, Brazil; sabrina.somacal@ufsm.br; 5Department of Technology and Food Science, Federal University of Santa Maria, Santa Maria 97105-900, RS, Brazil; tatiana.emanuelli@ufsm.br; 6Federal Institute of Pará, Bragança Campus, Computational Chemistry and Modeling Laboratory, Bragança 68600-000, PA, Brazil; joao.rocha@ifpa.edu.br

**Keywords:** flavonoids, inflammation, natural products

## Abstract

**Simple Summary:**

This paper reports the interaction between flavonoids, identified in the chemical matrix of açaí extract, and NLRP3 PYD through computational simulation, as well as the in vitro safety profile and anti-inflammatory effect in macrophages and monocytes of three flavonoids, isolated and combined, via the modulation of the NLRP3 inflammasome.

**Abstract:**

Inflammation is a vital mechanism that defends the organism against infections and restores homeostasis. However, when inflammation becomes uncontrolled, it leads to chronic inflammation. The NLRP3 inflammasome is crucial in chronic inflammatory responses and has become a focal point in research for new anti-inflammatory therapies. Flavonoids like catechin, apigenin, and epicatechin are known for their bioactive properties (antioxidant, anti-inflammatory, etc.), but the mechanisms behind their anti-inflammatory actions remain unclear. This study aimed to explore the ability of various flavonoids (isolated and combined) to modulate the NLRP3 inflammasome using in silico and in vitro models. Computer simulations, such as molecular docking, molecular dynamics, and MM/GBSA calculations examined the interactions between bioactive molecules and NLRP3 PYD. THP1 cells were treated with LPS + nigericin to activate NLRP3, followed by flavonoid treatment at different concentrations. THP1-derived macrophages were also treated following NLRP3 activation protocols. The assays included colorimetric, fluorometric, microscopic, and molecular techniques. The results showed that catechin, apigenin, and epicatechin had high binding affinity to NLRP3 PYD, similar to the known NLRP3 inhibitor MCC950. These flavonoids, particularly at 1 µg/mL, 0.1 µg/mL, and 0.01 µg/mL, respectively, significantly reduced LPS + nigericin effects in both cell types and decreased pro-inflammatory cytokine, caspase-1, and NLRP3 gene expression, suggesting their potential as anti-inflammatory agents through NLRP3 modulation.

## 1. Introduction

The inflammatory response is the activation of the immune system triggered by pathogenic agents or injury to cells and tissues. Inflammation plays a crucial role in protecting the organism from infections and restoring tissue homeostasis [1].

There are two types of inflammation: acute and chronic. Acute inflammation is self-limited, essential for the elimination of pathological agents and/or tissue repair. It is characterized by vascular leakage, a massive recruitment of leukocytes, such as neutrophils, the release of pro-inflammatory mediators, and the generation of reactive oxygen species (ROS) [2]. The inflammation attenuation begins with (i) neutrophil apoptosis; (ii) monocyte infiltration and macrophage maturation to remove apoptotic cells and cellular debris; (iii) the induction of anti-inflammatory mechanisms; (iv) the restoration of vascular permeability; and (v) the recovery of normal tissue function [3].

Chronic inflammation results from a persistent imbalance between pro-inflammatory mechanisms and pathways that facilitate the resolution of inflammation [4]. The chronic nature of inflammation stems from the failure and inefficiency of pathways that are normally responsible for resolving inflammation and restoring tissue homeostasis [5]. In chronic inflammatory conditions, the greatest damage to the host is often caused by the inflammatory response itself, rather than by any infection. Typically, chronic inflammation involves the excessive production of ROS, leading to oxidative stress, and an increased release of pro-inflammatory cytokines such as interferon-gamma (IFN-γ), tumor necrosis factor-alpha (TNF-α), interleukin 1-beta (IL-1β), and interleukin-6 (IL-6) [4,6].

Another mechanism that leads to pro-cytokine release is the NOD-like receptor family, pyrin domain containing 3 (NLRP3) inflammasome [7]. This inflammasome is a protein multicomplex that includes the protein NLRP3, an apoptosis-associated adapter protein containing a CARD domain (ASC) and pro-caspase-1. As soon as the NLRP3 protein is activated via some mediating agent, the pentameric or heptameric complex is formed by the three main components [8]. Therefore, the inflammasome acts as a pro-inflammatory inducer via the release of activated caspase-1. Activated caspase-1 causes the overproduction of IL-1β and IL-18, which consequently induce the expression of other pro-inflammatory cytokines, such as TNF-α, IL-6, and IFN-γ [9], characterizing a process called a “storm” of cytokines [10,11]. It is important to mention that NLRP3 is involved in several previously described inflammatory diseases, such as gouty arthritis [12], inflammatory bowel disease (IBD) [13], neuropsychiatric disorders [14,15,16], some autoimmune diseases, and even COVID-19 [17].

In this context, NLRP3 seems to be a central point of scientific investigation to develop new anti-inflammatory alternatives. Many natural health products (NHPs) have been evaluated for their anti-inflammatory potential. One example is *Euterpe oleracea* Mart., popularly known as açaí. Our group has already shown that açaí hydroalcoholic extract can act as an anti-inflammatory agent via NLRP3 inflammasome modulation, and it seems to be related to its chemical matrix [18,19,20,21,22]. However, the use of NHPs presents some limitations, including (i) a low bioavailability when consumed via oral administration; (ii) a low solubility due to their complex chemical matrix; and (iii) a sensibility to environmental exposures (temperature, oxygen, light, etc.); in addition, (iv) there are no clear policies in terms of the use of NHPs as a therapeutic alternative [23].

On the other hand, numerous studies have demonstrated the bioactive properties of isolated molecules, prompting our research group to explore their potential for pharmacological study development. Accordingly, the aim of this study was to investigate, both in silico and in vitro, whether the most concentrated molecules from the chemical matrix of açaí extract—both isolated and combined—can modulate the NLRP3 inflammasome.

## 2. Materials and Methods

### 2.1. Experimental Design

This is a computational and experimental study where the potential anti-inflammatory effect, via the NLRP3 inflammasome modulation, of the most concentrated molecules found in an açaí hydroalcoholic extract were evaluated. This included the isolated and synergic actions of the molecules. All the experiments described here were carried out in triplicate and on three independent days.

### 2.2. Freeze-Dried Hydroalcoholic Açaí Extract: Production and Characterization

The freeze-dried hydroalcoholic açaí extract was produced according to the method of de Souza [21].

For the quantification of polyphenols present in the chemical matrix, the lyophilized extract was purified by solid-phase extraction using C-18 reverse-phase cartridges (SPE-C18 cartridges, Strata C18-E, Phenomenex), as previously described [24], with the same modifications reported by Bochi et al. [25]. Non-anthocyanin phenolics purified fractions from *Euterpe oleracea* were analyzed by HPLC with a photodiode array (PDA) detector using a reverse-phase C-18 Hypersil Gold column (5 µm particle size, 150 mm, 4.6 mm) following the validated method described by Quatrin et al. [26]. The injection volume was 20 µL, and the mobile phases were composed of 5% (*v*/*v*) methanol in acidified water (0.1% (*v*/*v*) of formic acid) and 0.1% (*v*/*v*) of formic acid in acetonitrile. Chromatograms for non-anthocyanin phenolic quantification purposes were obtained at 280 nm for hydroxybenzoates, at 320 nm for hydroxycinnamates, and at 360 nm for flavonoids. The phenolic compounds from samples were identified by comparison with the retention time of authentic standards and the spectral data obtained from UV–visible absorption spectra. Stock solutions of standards references was prepared in the initial mobile phase and were diluted in equidistant points within the concentration range of the limit of quantification (LOQ)—60 mg/L. Calibration curve for gallic acid: y = 79089x + 81326 (r = 0.9984; protocatechuic acid: y = 41570x + 28970 (r = 0.9977); syringic acid: y = 82930x + 40566 (r = 0.9995); caffeic acid: y = 159186x + 120861 (r = 0.9995); *trans*-ferulic acid: y = 165905x + 113049 (r = 0.9965); catechin: y = 19861x + 21544 (r = 0.9975); kaempferol 3-*O*-*β*-D-glucopyranoside: y = 57482x − 95671 (r = 0.9969); apigenin: y = 53524x − 41979 (r = 0.9786), and luteolin: y = 64434x − 373423 (r = 0.8945). The limit of detection (LOD) and LOQ for gallic acid, 4-hydroxybenzoic acid, protocatechuic acid, syringic acid, caffeic acid, vanillic acid, *p*-coumaric acid, *trans*-ferulic acid, kaempferol 3-*O*-*β*-D-glucopyranoside, apigenin, and luteolin were, respectively, 0.012 and 0.037 ppm; 0.027 and 0.083 ppm; 0.008 and 0.024 ppm; 0.006 and 0.017 ppm; 0.011 and 0.033 ppm; 0.026 and 0.078 ppm; 0.028 and 0.084 ppm; 0.016 and 0.047 ppm; and 0.146 and 0.444 ppm. Compounds that are a derivative of one of the standard monomers were quantified by equivalence, and the results were expressed as mg per 100 g of dry sample weight.

For the anthocyanin quantification (purified fraction), samples were injected in a reverse-phase column C-18 Core–Shell Kinetex column (2.6 µm particle size, 100 mm, 4.6 mm) at 38 °C following the validated method described by Silva et al. [27]. The injection volume was 20 µL, and the mobile phases were composed of a solution of 3% of formic acid in water (*v*/*v*) and 100% acetonitrile at a flow rate of 0.9 mL/min. Chromatograms were obtained at 520 nm for quantification purposes. Cyanidin and peonidin derivatives were identified based on the order of elution and absorption spectrum [28]. Anthocyanins were quantified by equivalence of cyanidin-3-O-glucoside using a stock solution of cyanidin-3-O-glucoside prepared in the initial mobile phase and diluted in equidistant points within the concentration range of LOQ—20 mg L^−1^ (curve: y= 2.85185x + 0.037, r = 0.9891, LOD: 0.012 ppm, and LOQ: 0.037 ppm). Results were expressed as mg per 100 g of lyophilized extract.

### 2.3. In Vitro Safety Profile of Freeze-Dried Hydroalcoholic Açaí Extract

Kidney epithelial cells (VERO cell line, ATCC^®^ CCL-81™) were obtained from the Rio de Janeiro Cell Bank (BCRJ, 0245, Rio de Janeiro, RJ, Brazil) and cultured using Dulbecco’s Modified Eagle Medium (DMEM) with 10% of fetal bovine serum (FBS) and 1% antibiotic (penicillin—100 U/mL—and streptomycin—100 mg/mL). Looking forward to investigating the in vitro safety profile of açaí extract, VERO cells were seeded at 1.5 × 10^5^ cells/mL in 96-well plates. Cells were exposed to a concentration curve of açaí extract (0.01–100 µg/mL) for 24, 48, and 72 h. After these periods of incubation, the extract’s effects were evaluated with cellular viability and proliferation indexes, the production of nitric oxide (NO), the total levels of ROS, and the release of double-stranded DNA (dsDNA). Additionally, the genotoxicity and the release of hemoglobin (in erythrocytes) were examined.

### 2.4. Computer Simulation

#### 2.4.1. Molecular Docking

A molecular docking simulation was performed based on the characterization of the açaí hydroalcoholic extract, as described in Section 2.3. Here, the five most concentrated molecules found as part of the açaí extract chemical matrix were selected: catechin (828.20 ± 5.49 mg/100 g of extract), apigenin (251.48 ± 6.79 mg/100 g of extract), epicatechin (178.94 ± 5.61 mg/100 g of extract), epigallocatechin (82.89 ± 1.30 mg/100 g of extract), and taxifolin (83.14 ± 4.17 mg/100 g of extract). In this regard, looking forward to investigating if the anti-inflammatory capacity of açaí extract could be due to its chemical composition, the effectiveness of these five molecules in inhibiting NLRP3 through molecular docking simulations was analyzed. The main objective was to identify the most potent inhibitor among these molecules. Furthermore, these computational simulations were conducted using two different parameters: pH 7.4 of a basic nature and pH 6.5 of an acidic nature to mimic certain inflammatory conditions.

To perform these analyses, the molecular docking software AutoDock Vina version 1.1.2 (AD-Vina) [29,30] was used and integrated with the AMDOCK tool [31]. AD-Vina is widely recognized and used in the scientific community to predict the interactions between a target and the ligands of interest. In this case, the ligand selected was the NLRP3 pyrin domain (PYD), which is the same area where MCC950, a well-known NLRP3 inhibitor, is capable of binding. MCC950 was used as the standard in this analysis. The structures used in this work were obtained from an online database. The NLRP3 PYD domain was obtained from the PDB Protein Data bank with ID PDBID: 2NAQ. The other molecules were obtained from the PUBCHEM database: catechin (PUBCHEM ID 9064), apigenin (PUBCHEM ID 5280443), epicatechin (PUBCHEM ID 72276), epigallocatechin (PUBCHEM ID 72277), taxifolin (PUBCHEM ID 439533), and MCC950 (PUBCHEM ID 9910393). It is important to mention that this protocol followed the guidelines established in the literature [32] and was conducted on a high-performance platform to ensure accurate and reliable results. The formed complexes (target–ligand) were evaluated and analyzed in terms of binding energy and Root Square Mean Deviation (RMSD).

Discovery Studio was also used to identify and visualize the protein binding sites with high precision. Discovery Studio uses structural and sequence data from target proteins, along with information from known ligands, to perform similarity calculations and find likely binding regions. The tool can calculate an affinity score between proteins and ligands, thus providing an estimate of the degree of interaction between them [33,34]. After identifying the regions with the highest binding potential in NLRP3, the subsequent steps involved preparing the protein and ligand for the molecular docking process. Prior to initiating docking, certain preliminary procedures were conducted with the target molecule, referred to as the receptor. Initially, it was crucial to identify and address the charges present within the molecules while also rectifying any unbound atoms to ensure structural stability. Additionally, the solvation of the surrounding medium also was considered, involving interactions with water molecules [35]. Regarding the ligands, torsion adjustments were made, enabling their structures to conform to various spatial orientations throughout the docking procedure. These preparatory stages play a vital role in generating more precise and dependable outcomes when analyzing the interactions between the receptor and ligands. Receptor and ligand preparation was performed using the AMDock Tools software version 1.1.2, which is a component of the AutoDock Suite. The software used for visualization and image generation was pymol.

#### 2.4.2. Molecular Dynamics Simulation

The H++ server [36] was used to determine the protonation states of amino acid residues at pH 7.4. The ligands (catechin, epicatechin, apigenin, and MCC950) underwent quantum mechanics (QM) optimization at the HF/6-31G* level using the Gaussian09 software [37]. Following this optimization, partial atomic charges for the docked compounds were calculated at the same QM level using the RESP method [38]. The system was then prepared using the tLEaP module from the AMBER package [39], where it was solvated in a rectangular box with periodic boundary conditions. Water molecules were added according to the TIP3P model [40,41], and counter ions were introduced to neutralize the system’s charges and achieve a physiological concentration of 0.15 M. This was performed to replicate physiological conditions, ensuring an appropriate ionic strength for obtaining more biologically relevant results.

For the parametrization of the ligands and the enzyme, the AMBER GAFF and AMBER ff14SB force fields [42] were used, respectively. Subsequently, the systems underwent four stages of energy minimization using the NCYC method, which combines the Steepest Descent and Conjugate Gradient algorithms [43]. Initially, half of the minimization steps were carried out with the Steepest Descent method, followed by the Conjugate Gradient method until the end of the optimization. The minimization process included, in sequence, solvent relaxation, protein hydrogen relaxation, the simultaneous relaxation of both protein and solvent hydrogens, and, finally, a general system minimization.

In the production phase, 100 ns simulations were conducted for each system under constant temperature and pressure conditions (NPT). The trajectories generated during this phase were later analyzed and used for binding free energy calculations.

#### 2.4.3. Generalized Born and Surface Area Continuum Solvation (MM/GBSA)

To clarify the binding affinity of each simulated ligand, as well as the inhibitor MCC950, with the protein, we performed binding free energy calculations using the MM/GBSA method [44], available in the AmberTools23 package [45]. The mathematical basis of this methodology has already been described in previous publications [46]. The analysis of binding free energy and its decomposition was carried out based on the last 10 ns of the trajectories generated by the molecular dynamics simulations.

### 2.5. Bioactive Isolated Molecules’ Analysis

#### 2.5.1. Bioactive Isolated Molecules’ In Vitro Safety Profile

To analyze the safety profile of the isolated molecules, the VERO cell line was used. VERO cells were cultivated, maintained, and tested as described in Section 2.3. However, in this case, the performed treatments were conducted with the isolated catechin, apigenin, or epicatechin (0.01–100 µg/mL).

#### 2.5.2. Bioactive Isolated Molecules’ Anti-Inflammatory Effect in Monocytes

Human monocytes (THP-1 cell line) (ATCC^®^ TIB-202^TM^) were purchased from the Rio de Janeiro Cell Bank (BCRJ, 0234, Rio de Janeiro, RJ, Brazil) These cells were cultured using RPMI-1640 medium containing 10 mM of HEPES, 10% of FBS, and 1% antibiotics (penicillin—100 U/mL—and streptomycin—100 mg/mL). Cells were kept in a CO_2_ incubator settled at 5% CO_2_ and 37 °C.

THP-1 cells were exposed to an NLRP3 inflammasome activation and inhibition protocol. For a detailed protocol, please refer to Zhou et al. [47]. Briefly, cells were exposed to 100 ng/mL of lipopolysaccharide (LPS) for 3 h, followed by exposure to 10 µM of nigericin for 1 h. MCC950, an NLRP3 inflammasome inhibitor, was then administered at 100 nM for 2 h. Concurrently, LPS- and nigericin-activated monocytes were treated with various concentrations of each molecule (ranging from 0.01 to 100 µg/mL) to assess their anti-inflammatory properties through NLRP3 inflammasome modulation. Following all treatments and incubation periods, cells were analyzed for cellular viability, NO levels, total ROS levels, and extracellular dsDNA release.

### 2.6. Combined Bioactive Molecules’ Analysis

#### 2.6.1. Combined Bioactive Molecules’ In Vitro Safety Profile

After identifying the optimal concentrations of each isolated molecule for maximal anti-inflammatory effect, these molecules were then combined to enhance their therapeutic potential. Bioactive molecules were combined as follows: (i) catechin + apigenin; (ii) catechin + epicatechin; (iii) apigenin + epicatechin; and (iv) catechin + apigenin + epicatechin. The in vitro safety profile was performed using the VERO cell line. Cells were cultured and maintained as described in Section 2.3. These cells were exposed to the combined molecules for 24, 48, and 72 h of incubation. At the end of the incubation periods, cells were evaluated for cellular viability and proliferation, the levels of NO, the total levels of ROS, and the release of dsDNA. The genotoxicity and the release of hemoglobin (in erythrocytes) were also measured.

#### 2.6.2. Combined Bioactive Molecules’ Anti-Inflammatory Effect in Macrophages

First, THP-1 cells were induced to macrophage differentiation by using 25 ng/mL of phorbol 12-myristate 13-acetate (PMA) [48]. Looking forward to confirming macrophage generation, cellular morphology change was measured (from circle to fusiform shape and attachment). After the complete differentiation of monocytes into macrophages, the NLRP3 inflammasome activation and inhibition protocol was conducted as described in Section 2.5.2. Bioactive molecules were combined as described in Section 2.6.1. After all the treatments and periods of incubation, cells were analyzed for cell viability, the levels of NO, the total levels of ROS, and the extracellular dsDNA levels. The gene expression of the cytokines, caspase-1 and NLRP3 was assessed with the best combination of the molecules.

### 2.7. Experimental Analysis

#### 2.7.1. Cellular Viability and Proliferation Evaluation

Cell viability (24 h of incubation) and proliferation (48 and 72 h of incubation—considering the cellular duplication rate) were evaluated using the 3-(4,5-dimethylthiazol-2-yl)-2,5-diphenyltetrazolium (MTT) bromide assay (Sigma-Aldrich-M2128; St. Louis, MO, USA) following the instructions previously described [49]. This colorimetric assay consists of the intracellular metabolization of MTT into formazan crystals by the mitochondrial enzymes of viable cells. The absorbance was read at 570 nm by spectrophotometry using a Synergy H1 plate reader (Biotek, Santa Clara, CA, USA).

#### 2.7.2. Determination of Indirect NO Levels

The determination of indirect NO levels was performed as described in Choi et al. [50]. This is a colorimetric assay based on the use of the Griess reagent to detect metabolic nitrate and nitrite in the sample. The absorbance was read at 540 nm by spectrophotometry using a Synergy H1 plate reader (Biotek, Santa Clara, CA, USA).

#### 2.7.3. Total Levels of ROS Measurement

Total ROS levels were measured using the DCFH-DA reagent, following the protocol described by Costa et al. [51]. This is a fluorimetric assay based on the metabolization of dichlorodihydrofluorescein-diacetate (DCFH-DA) to dichlorodihydrofluorescein (DCFH) by intracellular enzymes. Upon contact with ROS, with greater sensitivity to hydrogen peroxide, DCFH is metabolized into dichlorofluorescein (DCF), and this molecule is capable of emitting fluorescence. Fluorescence levels were determined on a Synergy H1 plate reader (Biotek, Santa Clara, CA, USA) at 488 nm excitation and 525 nm emission.

#### 2.7.4. Extracellular dsDNA Measurement

Extracellular dsDNA quantification was performed using the Quant-iTTM PicoGreen^®^ reagent (Thermo Fisher-P11495; São Paulo, SP, Brazil) in the cell supernatant following Ahn, Costa, and Emanuel [52]. PicoGreen intercalates into dsDNA and emits fluorescence, making it possible to measure cellular integrity. Considering that when cells suffer membrane damage the dsDNA is released to the extracellular environment, this measurement could reflect an index of cellular mortality. Fluorescence was determined at 480 nm excitation and 520 nm emission in a Synergy H1 plate reader (Biotek, Santa Clara, CA, USA).

#### 2.7.5. Genomodifier Capacity Assay—GEMO

The GEMO assay is a non-cellular method that was carried out to identify the genomodifying (genotoxic and/or genoprotective) capacity of all molecules and extracts. The assay was conducted according to the instructions described by Cadoná et al. [53]. Calf thymus dsDNA was used as a standard sample and exposed to different concentrations of the molecules. Hydrogen peroxide was used as a pro-oxidant to cause damage to dsDNA to evaluate the genoprotective capacity. PicoGreen reagent was added, and fluorescence was emitted according to the concentration of intact dsDNA. Fluorescence was determined at 480 nm excitation and 520 nm emission in a Synergy H1 plate reader (Biotek, Santa Clara, CA, USA).

#### 2.7.6. Hemolysis Assay

The hemolysis test was carried out to verify whether red blood cells rupture as a result of exposure to individual, combined, and nanostructured molecules by detecting the rate of hemoglobin release. For this, heparinized peripheral blood was collected, and the red blood cells were washed with 1× PBS buffer (1:1) and centrifuged at 190× *g* for 5 min each wash. Afterwards, 400 µL of red blood cells, 1 mL of 1× PBS, and 80 µL of treatment were added to falcon tubes and incubated at 37 °C for 1 h. For the positive control of hemolysis, distilled water was used. At the end of incubation, the tubes were centrifuged at 190× *g* for 5 min, and 100 µL of the supernatant was transferred to a 96-well plate. Absorbance was analyzed at 409 nm using a Synergy H1 plate reader (Biotek, Santa Clara, CA, USA). Human research ethics committee approval: 31211214.4.0000.5306.

#### 2.7.7. Cytokine, Caspase-1, and NLRP3 Gene Expression

After determining the most effective concentration of molecules in decreasing NLRP3 activation, the gene expressions of IL-1β, IL-6, TNF-α, IL-10, caspase-1, and the NLRP3 inflammasome were measured following the descriptions of the manufacturers.

RNA was extracted by using TRI Reagent^®^ (Sigma-Aldrich, Saint Louis, MO, USA) and was quantified using NanoDrop Lite^®^ (ThermoFisher Scientific, Wilmington, DE, USA) equipment. The RNA amount was normalized for each sample at a final concentration of 100 ng. Complementary DNA (cDNA) was produced by using the iScript™ cDNA Synthesis kit (Bio-Rad, Hercules, CA, USA) following the manufacturer’s instructions. Quantitative real-time PCR was carried out using a GoTaq^®^ qPCR Master kit (Promega, Madinson, WI, USA). qRT-PCR cycles were as follows: (1) 50 °C for 120 s; (2) 95 °C for 120 s; and (3) 40 cycles of 95 °C for 15 s followed by 60 °C for 30 min. IL-1β primers were as follows: forward—5′ AAGCCCTTGCTGTAGTGGTG 3′; reverse—5′ GAAGCTGATGGCCCTAAACA 3′. IL-6 primers were as follows: forward—5′ AGACAGCCACTCACCTCTTCAG 3′; reverse—5′ TTCTGCCAGTGCCTCTTTGCTG 3′. TNF-α primers were as follows: forward—5′ CTCTTCTGCCTGCTGCACTTTG 3′; reverse—5′ ATGGGCTACAGGCTTGTCACTC 3′. IL-10 primers were as follows: forward—5′ GTGATGCCCCAAGCTGAGA 3′; reverse—5′ TGCTCTTGTTTTCACAGGGAAGA 3′. NLRP3 primers were as follows: forward—5′ CCCCGTGAGTCCCATTA 3′; reverse—5′ GACGCCCAGTCCAACAT 3′. Caspase-1 primers were as follows: forward—5′ CGCACACGTCTTGCTCTCAT 3′; reverse—5′ TACGCTGTACCCCAGATTTTGTAG 3′. Beta-actin was the housekeeping gene. Beta-actin primers were as follows: forward—5′ CTGGCACCACACCTTCTAC 3′; reverse: 5′-GGGCACAGTGTGGGTGAC 3′.

#### 2.7.8. Measurement of Lactate Levels

Lactate levels were measured on an automated biochemical analyzer using a commercial lactate enzyme kit following the manufacturer’s instructions (Labtest Diagnóstica S.A., Lagoa Santa, MG, Brazil).

### 2.8. Statistical Analysis

The obtained results were converted to percentage ± standard deviation related to the untreated cells group (negative control). Then, statistical analysis was performed using GraphPad Prism 8.0.1 (GraphPad Prism^Ⓡ^, 2018; San Diego, CA, USA) software by one-way ANOVA, followed by Tukey post hoc. *p* < 0.05 was considered statistically significant. The data obtained for gene expression were analyzed using the delta–delta Ct calculation and relative gene expression conversion, with these results being compared with the control group.

## 3. Results

### 3.1. Production and Characterization of Açaí Extract

A homogeneous, purple-colored, and freeze-dried hydroalcoholic açaí extract was obtained and was subsequently characterized by HPLC. Eighteen bioactive molecules were found in the extract, specifically, gallic acid (peak 1; 1.62 ± 0.13 mg/100 g), protocatechuic acid (peak 2; 8.87 ± 1.77 mg/100 g), epigallocatechin (peak 3; 82.89 ± 1.30 mg/100 g), catechin (peak 4; 838.20 ± 5.49 mg/100 g), syringic acid (peak 5; quantification below the limit of the analytical method), epicatechin (peak 6; 178.94 ± 5.61 mg/100 g), taxifolin (peak 7; 83.14 ± 4.17 mg/100 g), t-cinnamic acid (peak 8; 0.09 ± 0.00 mg/100 g), caffeic acid (peak 9; 24.66 ± 2.77 mg/100 g), t-ferulic acid (peak 10; 6.15 ± 0.91 mg/100 g), apigenin (peak 11; 251.48 ± 6.79 mg/100 g), orienthin (peak 12; 27.69 ± 0.80 mg/100 g), kaempferol 3-β-D-glucopyranoside (peak 13; 82.89 ± 1.30 mg/100 g), luteolin (peak 14; 53.83 ± 0.44 mg/100 g), cyanidin-3-O-glucoside (peak 15; 15.51 ± 1.31 mg/100 g), cyanidin-3-O-rutinoside (peak 16; 16.08 ± 0.64 mg/100 g), peonidin-3-O-glucoside (peak 17; 0.44 ± 0.02 mg/100 g), and peonidin-3-O-rutinoside (peak 18; 0.27 ± 0.09 mg/100 g) (Figure 1).

### 3.2. In Vitro Safety Profile of Açaí Extract

Different açaí extract concentrations were tested to see if there were any cellular modulations in VERO cells by themselves (per se effect) (Figure 2). Most of the tested açaí extract concentrations did not cause any significant modifications in the cells after 24 and 72 h of incubation (Figure 2A–D,I–L). However, after 48 h of incubation, most concentrations of açaí extract increased the cell proliferation index (Figure 2E), while the concentrations of 0.1, 1, and 100 µg/mL increased the release of extracellular dsDNA (Figure 2H) when compared to the control group; the NO levels remained unchanged (Figure 2F), and there was a significant decrease in the ROS rate under the concentrations of 0.1 and 100 µg/mL (Figure 2G) in comparison to untreated cells. In the genomodulation assay, none of the concentrations showed a genotoxic effect (Appendix A), and only the concentration of 100 µg/mL of açaí extract demonstrated a genoprotective effect when compared to the positive control (Appendix A). Regarding the hemolysis assay, none of the tested concentrations of açaí extract caused a hemolytic effect in red blood cells (Appendix A).

### 3.3. Molecular Docking

The molecular docking of bioactive molecules (ligands) was performed with the PYD region of the NLRP3 inflammasome (target). Table 1 shows the results of the calculated binding affinities and inter-atom distance (RMSD) of each ligand with the target with pH 7.4 and 6.5. The analysis of the molecular docking results showed that MCC950, a known inhibitor of the NLRP3 inflammasome, has an important binding affinity (−7.5 kcal/mol) with the NLRP3 PYD domain. On the other hand, mainly catechin, apigenin, and epicatechin obtained binding affinity results (−7.52, −7.1, and −6.3 kcal/mol, respectively) with the same domain close to that of MCC950. Furthermore, all ligands evaluated showed satisfactory RMSD < 2. The results found for the molecular docking between the bioactive molecules and the PYD domain of NLRP3 at pH 6.5 were very similar to those described with pH 7.4.

Then, the bioactive molecules chosen to generate the 3D graph and 2D map were catechin, apigenin, and epicatechin.

Figure 3A–D,I–L show a 3D graph of the positions of interactions between catechin, apigenin, epicatechin, and MCC950, respectively, and the NLRP3 PYD domain. The 2D maps show the possibilities of interaction and the type of binding of each molecule with the amino acids of the PYD domain of NLRP3. For interactions carried out at pH 7.4, the amino acid residues of the interaction between catechin and NLRP3 are Glu, Pro, Arg, Gln, Gly, Leu, Tyr, and Cys through van der Waals, conventional hydrogen, carbon hydrogen, amide-Pi stacked, and Pi-alkyl bonds. (Figure 3E). Apigenin and NLRP3 present the amino acid residues Glu, Pro, Gln, Gly, Leu, Tyr, Cys, Ile, and Asp through van der Waals, conventional hydrogen, Pi-anion, Pi-sulfur, PI-lone pair, Pi-Pi T-shaped, and Pi-alkyl bonds (Figure 3F). Epicatechin and NLRP3 present the amino acid residues Glu, Pro, Leu, Asp, Ile, and Cys through van der Waals, conventional hydrogen, Pi-donor hydrogen, Pi-Sigma, Pi-sulfur, and Pi-alkyl bonds and an unfavorable donor–donor with Tyr (Figure 3G). MCC950 and NLRP3 present the amino acid residues Glu, Pro, Lys, Ala, Tyr, Trp, and Val through van der Waals, conventional hydrogen, Pi-Sigma, Pi-Pi stacked, and alkyl bonds (Figure 3H). Although the binding affinities between the ligands and the target were similar at pH 7.4 and 6.5, some interactions and amino acid residues were distinct. Therefore, the amino acid residues at pH 6.5 for the interaction between catechin and NLRP3 were Glu, Pro, Leu, Ile, Gly, Gln, Cys, and Asp through van der Waals, conventional hydrogen, carbon hydrogen, Pi-donor hydrogen, Pi-Sigma, Pi-sulfur, and Pi-alkyl bonds and an unfavorable donor–donor with Tyr (Figure 3M). Apigenin and epicatechin present the same amino acid residues and interactions of the simulations with pH 7.4 (Figure 3N,O). Finally, MCC950 and NLRP3 present the amino acid residues Glu, Ala, Arg, Trp, Tyr, Lys, and Val through van der Waals, conventional hydrogen, Pi-Sigma, Pi-sulfur, Pi-Pi stacked, alkyl, and Pi-alkyl bonds (Figure 3P).

### 3.4. Molecular Dynamics Simulations

Molecular dynamics simulations were carried out for 100 ns to evaluate the structural stability of the complexes formed between the PYD domain of the NLRP3 inflammasome and the ligands catechin, apigenin, epicatechin, and MCC950. Figure 4 displays the RMSD profiles over time.

The docking results demonstrated that catechin, apigenin, and epicatechin exhibit binding affinities comparable to the reference inhibitor MCC950, with binding energy values near −7.5 kcal/mol. However, upon analysis of molecular dynamics simulations, the MCC950 complex exhibited greater structural variation, with RMSD values ranging from 4 to 5 Å, indicating greater conformational flexibility. This suggests that, despite the strong affinity observed in docking, MCC950 may induce conformational changes in NLRP3 during binding, corroborating its enhanced inhibitory capacity. Similar studies have also identified structural fluctuations induced by high-affinity inhibitors of NLRP3 [54,55].

Figure 5 presents the residual fluctuations along the NLRP3 amino acid sequence. Regions with greater fluctuation (Val18 to Lys22) presented by the catechin and apigenin complexes suggest the lower stabilization of interactions with the PYD domain, consistent with docking values that indicated slightly lower affinity when compared to MCC950. In contrast, epicatechin and MCC950 exhibited lower fluctuations in critical residues between Leu20 and Pro40, suggesting more stable interactions with the inflammasome.

These reduced residual fluctuations may be associated with greater conformational rigidity, indicating that these ligands stabilize the protein more effectively. This aligns with the literature, which suggests that ligands with higher affinity generally result in greater conformational stabilization [56]. The reduced fluctuations in critical residues for inflammasome function, such as Pro38 and Leu39, also suggest the enhanced inhibition of NLRP3 activation by epicatechin and MCC950. Moreover, these results are consistent with docking findings, where MCC950 and epicatechin exhibited favorable interactions with residues such as Glu and Pro in the PYD domain, which are essential for inflammasome activation and assembly.

Table 2 presents solvent-accessible surface area (SASA) values for the different complexes. MCC950 exhibited the highest SASA (1155.29 Å^2^), suggesting that the binding of this compound causes greater protein exposure to the solvent, potentially related to the conformational changes observed in RMSD. Epicatechin, with a SASA of 1093.74 Å^2^, exhibited a behavior similar to that of MCC950, suggesting that both ligands induce a more open conformation of the inflammasome. These SASA results align with docking interactions, where MCC950 and epicatechin presented multiple interactions with polar and non-polar residues on the protein surface, potentially resulting in greater exposure to the solvent.

In contrast, apigenin resulted in the lowest solvent exposure (1027.80 Å^2^), suggesting that this compound induces a more compact protein conformation, possibly related to the lower structural variation observed in RMSD. This observed compactness may indicate more effective inhibition of the inflammasome, as a reduction in SASA is frequently associated with more stable and less solvent-accessible complexes [57].

The Rg values indicate that apigenin induced the highest protein compactness, with a value of 11.79 Å, confirming the SASA results. Catechin and epicatechin exhibited slightly higher values (11.88 Å and 11.98 Å, respectively), with epicatechin displaying a behavior similar to that of MCC950 (11.97 Å). These results suggest that epicatechin and MCC950 induce a more open protein conformation while maintaining a certain degree of structural stability, whereas apigenin favors a more compact conformation, which may influence the efficiency of inflammasome inhibition.

RMSD and Rg analyses suggest that the simulations reached equilibrium after approximately 50 ns, with values stabilizing by the end of the 100 ns simulations. This indicates that the systems reached a stable conformation and that the data collected from the simulations are representative of equilibrated states. This convergence has been observed in similar molecular dynamics studies, which have demonstrated that simulation times of 100 ns are sufficient to stabilize ligand–protein complexes [58,59].

The free energy and energy decomposition results for the ligand–protein complexes will be discussed in the following sections, providing a more detailed analysis of the key interactions stabilizing the complexes.

### 3.5. MM/GBSA Binding Free Energy Calculations

To estimate the binding affinity and inhibitory potential of apigenin, catechin, epicatechin, and the inhibitor MCC950 in relation to the PYD domain of NLRP3 in the protein–ligand complexes, binding free energy calculations were performed. The results obtained from the MM/GBSA method are presented in Table 3.

Table 3 shows the binding free energy components calculated using the MM/GBSA method for the complexes formed between the PYD domain of NLRP3 and the ligands apigenin, catechin, epicatechin, and MCC950. The values include contributions from van der Waals energy (∆E_vdW_), electrostatic energy (∆_Eele_), polar solvation energy (∆G_GB_), and non-polar solvation energy (∆G_nonpol_). The total binding free energy (∆G_MM/GBSA_) is also presented for each molecule, with MCC950 showing the highest binding affinity, followed by epicatechin, apigenin, and catechin.

The energy decomposition per residue, illustrated in Figure 6, reveals the main energetic contributions of each residue involved in the interaction between the PYD domain of NLRP3 and the ligands MCC950, apigenin, catechin, and epicatechin. For MCC950 (∆G_MM/GBSA_ = −30.6931 kcal/mol), residues such as Gln33, Pro32, Pro38, and Leu39 contributed significantly to the stabilization of the complex, with energy contributions lower than −2.00 kcal/mol, suggesting a strong interaction in these critical regions.

Similarly, for apigenin (∆G_MM/GBSA_ = −7.1022 kcal/mol), residues Pro31, Pro32, Cys36, and Pro38 were identified as important contributors to the stabilization of the complex, which is consistent with the binding affinity observed in the free energy calculations. In contrast, the catechin complexes (∆G_MM/GBSA_ = −6.3203 kcal/mol) showed more relevant energetic contributions from residues such as Lys21, Lys22, and Val18, suggesting a different interaction pattern compared to the other ligands.

For the epicatechin complex (∆G_MM/GBSA_ = −21.1365 kcal/mol), the energy decomposition highlighted residues Cys36, Pro38, Leu39, and Arg41 as key stabilizers, reinforcing the importance of these interactions in binding to the NLRP3 inflammasome. These results are consistent with the binding free energy analysis previously presented, where epicatechin showed significant affinity.

In summary, the energy decomposition analyses provide a detailed view of the key residues involved in the stabilization of each protein–ligand complex, highlighting the relevance of residues such as Cys36, Pro38, and Leu39 in the inhibition of the NLRP3 inflammasome, especially for the natural ligands derived from açaí. This further suggests that these natural compounds bind differently compared to the MCC950 inhibitor.

### 3.6. In Vitro Safety Profile of Bioactive Isolated Molecules

The three main molecules that had the highest binding affinity with the PYD NLRP3 inflammasome domain through molecular docking were chosen to conduct the in vitro assays: catechin, apigenin, and epicatechin. All isolated molecules were tested for their in vitro safety profile in VERO cells after 24, 48, and 72 h of incubation.

VERO cells exposed to catechin did not have significant changes in cell viability (Figure 7A) or the NO (Figure 7B) and dsDNA levels in the extracellular medium (Figure 7D) compared to untreated cells. In contrast, all tested concentrations of catechin were able to reduce ROS levels in relation to the negative control after 24 h of incubation (Figure 7C). In the GEMO assay, there was no genotoxic activity (Appendix A), and it presented genoprotective properties (Appendix A) at the highest concentrations tested in comparison to the positive control.

Most of the apigenin concentrations increased cell viability rates after 24h of exposure when compared to the negative control (Figure 7E). Other parameters did not show significant changes in relation to untreated cells (Figure 7F–H). Apigenin was not genotoxic (Appendix A), and the concentrations of 10 and 100 µg/mL had genoprotective capacity when compared to the H_2_O_2_ positive control (Appendix A).

Most parameters assessed in VERO cells following the exposure to epicatechin remained stable across most tested concentrations after 24 h of incubation compared to the negative control (Figure 7I–L). Epicatechin did not demonstrate genotoxicity (Appendix A), and it was genoprotective at all concentrations tested through the GEMO assay (Appendix A).

After 48 and 72 h of exposure to each isolated bioactive molecule, the cellular parameters evaluated remained similar to those observed after 24 h of incubation (Appendix A).

Catechin (Appendix A), apigenin (Appendix A), and epicatechin (Appendix A) did not demonstrate a hemolytic capacity.

### 3.7. Anti-Inflammatory Effect of Bioactive Isolated Molecules

The exposure of THP-1 cells to LPS + nigericin resulted in reduced cell viability and elevated levels of NO, ROS, and extracellular dsDNA, compared to the non-activated control cells. Treatment with MCC950 and various concentrations of each bioactive molecule demonstrated a reduction in these inflammatory markers relative to the LPS + nigericin positive control, as shown in Figure 8. Specifically, catechin at a concentration of 1 µg/mL was most effective in promoting cellular recovery. Similarly, apigenin at 0.1 µg/mL not only enhanced cellular viability but also reduced markers of oxidative stress. Furthermore, a dose of 0.01 µg/mL of epicatechin achieved comparable effects in mitigating the inflammatory response induced by LPS + nigericin.

### 3.8. In Vitro Safety Profile of Bioactive Combined Molecules

After determining the best concentration of each isolated bioactive molecule capable of reversing the effects induced by *LPS + nigericin* in monocytes, these molecules were combined and evaluated for their in vitro safety profile. Therefore, the concentrations chosen were as follows: (i) catechin 1 µg/mL; (ii) apigenin 0.1 µg/mL; and (iii) epicatechin 0.01 µg/mL. The combinations used were as follows: (i) catechin + apigenin; (ii) catechin + epicatechin; (iii) apigenin + epicatechin; and (iv) catechin + apigenin + epicatechin.

VERO cells were exposed to the combined bioactive molecules for 24, 48, and 72 h. There were no significant changes in any incubation time following exposure to the combined bioactive molecules compared to the negative control, suggesting that all combinations have a desirable in vitro safety profile in cells (Figure 9A–L). Additionally, the combined bioactive molecules did not show a genotoxic effect (Figure 9M) or hemolytic capacity (Figure 9N).

### 3.9. Anti-Inflammatory Effect of Bioactive Combined Molecules in THP-1-Derived Macrophages

Initially, THP-1 cells were treated with PMA to induce their differentiation into macrophages. PMA exposure induces monocytes (Figure 10A) to adhere to the cell culture flask and transition from a circular to a fusiform shape, as depicted in Figure 10B. When activated by LPS + nigericin, these macrophages exhibited decreased cell viability (Figure 10C) and increased levels of NO (Figure 10D), ROS (Figure 10E), and extracellular dsDNA (Figure 10F) compared to untreated cells. Treatment with MCC950 and combinations of bioactive molecules improved cell viability and reduced the levels of NO and dsDNA. However, ROS levels remained unchanged across all treatments.

After choosing the best combination of bioactive molecules capable of reversing the effects caused by LPS + nigericin, the ability of such isolated and combined molecules to modulate the gene expression of caspase-1, NLRP3, IL-1β, IL-6, TNF-α, and IL-10 was analyzed. Figure 11A shows the cellular morphology of each condition after treatment with the chosen combination. It is possible to observe that the LPS + nigericin group had an expressive change in terms of cellular number and morphology compared to untreated cells. On the other hand, MCC950 as well as all the combinations tested for the bioactive molecules could keep cellular shape in a similar way to the negative control.

For lactate, there was a decrease in lactate dosage in the positive control LPS + nigericin in relation to the negative control. Macrophages treated with MCC950 and epicatechin showed decreased lactate levels, while catechin showed elevated lactate levels compared to the LPS + nigericin group (Figure 11B). Macrophages exposed to LPS + nigericin showed a significantly higher differential gene expression of caspase-1, NLRP3, IL-6, and TNF-α compared to untreated cells (Figure 11C,D,F,G, respectively). Cells exposed to LPS + nigericin also present higher levels of IL-1β; however, this result was not significant (Figure 11E). Regarding IL-10 (Figure 11H), macrophages exposed to LPS + nigericin did not present significant changes compared to the negative control. Macrophages activated by LPS + nigericin and treated with isolated MCC950, catechin, and epicatechin showed a reduced gene expression of caspase-1, NLRP3, IL-6, and TNF-α compared to the LPS + nigericin positive control. Despite LPS + nigericin exposure not increasing IL-1β levels, catechin and epicatechin levels were found to reduce the gene expression of this cytokine in comparison to the positive control. On the other hand, treatment with the combined molecules was able to reduce the gene expression of caspase-1 and TNF-α but not NLRP3, IL-1β, or IL-6. A significant increase in the expression of NLPR3 was observed under catechin + epicatechin treatment.

## 4. Discussion

The therapeutic benefits attributed to natural health products (NHPs) have been well documented for decades, sparking increasing scientific interest due to their potential in treating various conditions. However, identifying the bioactive molecules responsible for these biological effects and understanding the physiological mechanisms involved is crucial, especially when NHPs are used for therapeutic purposes. Consequently, this study evaluated the potential anti-inflammatory effects of isolated and combined bioactive molecules from the freeze-dried hydroalcoholic extract of açaí. These effects were assessed through the modulation of the NLRP3 inflammasome using both in silico and in vitro models.

Firstly, the açaí extract was produced and characterized. The bioactive molecules found with the highest content in the quantification of the chemical matrix of the açaí extract were catechin, apigenin, epicatechin, taxifolin, and epigallocatechin. All of these polyphenols are classified as flavonoids. Flavonoids have similar biological effects despite their chemical differences, such as anti-carcinogenic [60,61], antioxidant [62], anti-inflammatory [63], and neuroprotective effects [64,65].

The in vitro safety profile of this extract was determined using a concentration curve from 0.01 to 100 μg/mL in VERO cells. No significant cytotoxic effects were observed after exposing VERO cells to all concentrations of açaí extract. This finding was also observed in studies by de Souza et al. [21] and Davidson et al. [22] in microglial and lung cells, respectively, after exposure to a similar concentration curve of açaí extract. All these results suggest that açaí extract has a desirable safety profile in vitro in different cell lines. Regarding the anti-inflammatory effect of açaí extract, Machado et al. [19] found that 1 μg/mL of açaí extract can modulate the NLRP3 inflammasome, reducing inflammatory effects in RAW 264.7 macrophages activated with phytohemagglutinin (PHA). Later, Cadoná et al. [20] investigated the potential anti-neuroinflammatory effect of açaí extract in microglia cells through the NLRP3 inflammasome priming and activation pathway both isolated and combined. In both studies, the authors suggest that the anti-inflammatory effect of açaí is attributed to the chemical matrix of this NHP and that the bioactive molecules have an isolated or synergistic effect.

By aiming to explore the anti-inflammatory capacity using NLRP3 inflammasome modulation of the açaí extract and its chemical matrix, molecular docking of the interaction of the five most abundant molecules found in the açaí extract was conducted in the NLRP3 inflammasome PYD domain. The interactions between the bioactive molecules and the PYD domain of NLRP3 were performed, simulating a neutral pH environment and a slightly acidic one. These parameters were used because under normal conditions, physiological pH is tightly maintained and regulated at the cellular, tissue, and systemic levels. However, altered pH, especially acidic pH, is associated with physiological and pathological inflammation, infections, tumors, and other disturbances in body homeostasis [66]. In general, this occurs because during inflammation immune cells infiltrate the tissue, resulting in increased oxygen and glucose demand, which increases lactic acid secretion and consequently the acidification of pH [67]. Additionally, activated macrophages play a central role in the pathophysiology and progression of many chronic inflammatory diseases, since they secrete large amounts of pro-inflammatory cytokines and chemokines, such as IL-1β and IL-18. The maturation and secretion of these cytokines are regulated by the NLRP3 inflammasome pathway, which under inflammatory conditions activates the pro-inflammatory cytokine cascade contributing to the progression and severity of inflammatory events [68]. The interactions between bioactive molecules and NLRP3 were found to not undergo significant changes with the change in pH, indicating that even under altered conditions such interactions can still occur satisfactorily. Although no pH modification tests were carried out in this study, this information will serve as a basis for future studies.

It is important to highlight that in silico studies are an important tool in screening therapeutic candidates, as they provide predictive data on the behavior of molecules with the desired target. The NLRP3 interaction region chosen was the PYD domain due to the large number of protein regions capable of forming bonds through interactions with regions of bioactive molecules. Also, the PYD region is crucial in the oligomerization and activation of the protein complex through its interaction with the ASC domain [69,70,71,72]. The results obtained here demonstrated that the three molecules with the highest binding affinity with the PYD region were, in ascending order, as follows: catechin, apigenin, and epicatechin. Fang et al. [73] performed a molecular docking study investigating the benefits of different flavonoids against the NLRP3 inflammasome focusing on cardiovascular diseases. The authors found very similar results for catechin, apigenin, epigallocatechin, and taxifolin, which supports our findings. It is important to mention that there are few studies on the interaction between the same flavonoids identified in the present study and the NLRP3 inflammasome, demonstrating the innovative aspect of this research. Therefore, the molecular docking results suggest that catechin, apigenin, and epicatechin can act as potential NLRP3 inhibitors based on the values found for MCC950. Therefore, these molecules (catechin, apigenin, and epicatechin) were chosen to conduct the in vitro assays.

The molecular dynamics simulations conducted over 100 ns indicated that the protein–ligand complex systems converged to stable conformations, as evidenced by the stabilization of RMSD values after approximately 50 ns. The RMSD of the proteins bound to natural compounds derived from açaí, such as apigenin, catechin, and epicatechin, showed less variation compared to the APO form, suggesting structural stabilization induced by the binding of the compounds to the NLRP3 PYD domain. Similarly, the RMSF indicated greater conformational rigidity in the critical binding regions, suggesting that the interaction with these compounds favors complex stability.

Additionally, the analysis of SASA revealed that the natural ligands reduced protein exposure to the solvent, indicating a more compact and less accessible conformation, while the Rg corroborated this structural compaction observed in the simulations. These data suggest that açaí-derived ligands stabilize more compact and inactive conformations of the NLRP3 inflammasome, similarly to the MCC950 inhibitor.

When comparing the docking results with the molecular dynamics simulations, we observed marked differences in the binding affinity of the studied compounds. Although docking provided an initial estimate of the binding affinity between the ligands and the NLRP3 PYD domain, the binding free energy calculations using MM/GBSA revealed a more accurate picture. Compounds such as MCC950 showed significantly lower (more negative) energies in the MM/GBSA calculations, reflecting a higher binding affinity compared to natural compounds such as catechin and apigenin.

These differences are expected, given that traditional docking provides only a static view of the protein–ligand interaction, whereas MM/GBSA calculations account for dynamic solvent effects, conformational fluctuations of the protein and ligand, and interactions over time. Previous studies also support this observation, suggesting that molecular dynamics simulations are more accurate in capturing binding stability in complex biological systems [74,75].

In analyzing energy decomposition per residue, critical residues for complex stabilization with MCC950 were identified, such as Gln33, Pro32, Pro38, and Leu39, with energy contributions greater than −2 kcal/mol. These residues were also observed in relevant interactions with other ligands, such as epicatechin and apigenin, although with lower energy contributions, which may justify the lower binding affinity of these natural compounds. The comparison with docking results showed that while the initial affinity seemed similar among the compounds, the MM/GBSA methods provided a more accurate assessment of the complex’s stability over time, demonstrating the superiority of MCC950. However, natural products, such as those derived from açaí, offer advantages over synthetic inhibitors like MCC950, including lower toxicity, better bioavailability, and fewer side effects, as highlighted by previous studies on the therapeutic potential of these natural compounds [58].

In this study, it is suggested that the integration of methods such as MM/GBSA into molecular dynamics simulations is crucial for assessing the effectiveness of inhibitors targeting the NLRP3 inflammasome, as they provide a more detailed analysis of dynamic interactions, especially in proteins with high conformational flexibility.

In terms of the molecules’ chemical structures, catechins are natural polyphenols (flavan-3-ol or flavanol) from the flavonoid family [76]. The chemical presentation of catechin consists of two benzene rings and a heterocyclic dihydropyran ring with a hydroxyl group on carbon 3. The stereoisomers of catechin in the cis configuration ((-)-epicatechin) or trans ((+)-catechin), in relation to carbons 2 and 3, are flavan-3-ol compounds [77]. Apigenin (4′-5-7-trihydroxyflavone) is a flavone that also belongs to the flavonoid family [78]. In nature, apigenin is mainly present in glycosylated form, and the central tricyclic structure is linked to a sugar through hydroxyl groups (O-glycosides) or directly to carbon (C-glycosides).

The bioactive molecules were investigated regarding their in vitro safety profile. Therefore, VERO cells were exposed to each of the isolated molecules at 24, 48, and 72 h with the same concentration curve used for the açaí extract for the purpose of comparison, equivalence, and consistency. No significant cytotoxic effects were observed after periods of exposure to the molecules; however, catechin demonstrated a significant decrease in ROS levels compared to untreated cells, especially after 24 h. This effect may be attributed to the intrinsic antioxidant characteristics of catechins in general. There are studies suggesting that catechins can eliminate free radicals [77,79,80]. The antioxidant effect of catechins occurs through (i) direct mechanisms by the elimination of ROS and chelation of metal ions and (ii) indirect mechanisms, inducing the action of antioxidant enzymes and the inhibition of pro-oxidant enzymes, suppressing oxidative stress factors [81]. Catechin and diastereoisomers have common chemical structures (phenolic hydroxyl groups) that are capable of stabilizing free radicals, as they can react with ROS and reactive nitrogen species (RNS) in a termination reaction, breaking the cycle of the generation of new radicals [82].

After determining the in vitro safety profile, each molecule was evaluated for its ability to reverse inflammatory parameters via modulation of the NLRP3 inflammasome. For this, an NLRP3 activation protocol was used with LPS as a priming signal and nigericin as a protein complex assembly signal. MCC950, a known synthetic NLRP3 inhibitor, was used as the inhibition control. We observed that most of the tested concentrations of bioactive molecules managed to reduce NO and ROS levels to levels equivalent to MCC950, suggesting that they can inhibit NLRP3 activation in a similar way to the known inhibitor.

To further support our findings, there are some studies that suggest the anti-inflammatory potential of these molecules via the modulation of the NLRP3 inflammasome [83,84,85,86]. Jhang et al. [83] found that catechin was able to suppress the release of IL-1β and reduce the activation of the NLRP3 inflammasome; in addition, catechin was able to prevent mitochondrial damage induced by gouty arthritis. Picciolo et al. [86] identified that a mixture of different types of catechins decreased the expression of nuclear factor (NF)-κB and interrupted the activity of the NLRP3 inflammasome, as well as the expression of IL-1β, IL-18, and caspase-1 in human gingival fibroblasts and oral mucosal epithelial cells activated by LPS. Prince et al. [84] found that epicatechin from the diet was able to attenuate oxidative stress and change NO metabolism, as well as reduce inflammatory levels in the renal cortex of rats fed with fructose. The authors emphasize that epicatechin could be beneficial for treating kidney inflammatory conditions. Additionally, Tian et al. [85] found that epicatechin reversed lung inflammation induced by cigarette smoke through the NLRP3 inflammasome pathway and by modulating oxidative stress in rats with chronic obstructive pulmonary disease (COPD). In a study by Li et al. [87], apigenin was shown to inhibit the expression of the NLRP3 inflammasome and, consequently, the cytokine cascade in a model of neuroinflammation caused by psychological stress in rats’ brain slices. Complementarily, the study developed by Martínez, Mijares, and Sanctis [88] showed that in addition to inhibiting the expression of the NLRP3 inflammasome and the cytokine cascade, apigenin was also able to reduce the in vitro generation of ROS and RNS.

With the intention of evaluating whether there is a synergic effect of the molecules, the flavonoids were combined based on the most effective concentration of each one tested in their isolated form in modulating the NLRP3 inflammasome. Therefore, the concentrations chosen were as follows: 1 µg/mL of catechin, 0.1 µg/mL of apigenin, and 0.01 µg/mL of epicatechin. Each molecule was combined in pairs, and a combination of the three molecules was also prepared. Regarding the in vitro safety profile, VERO cells did not suffer significant cytotoxic effects. However, despite the combined molecules not being able to cause cytotoxic damage or hemolytic activity, the combination of the three molecules caused genotoxic damage found through the GEMO assay. This effect may be associated with the chemical structure of the flavonoids, such as the free hydroxyl groups that can oxidize depending on the pH of the environment, especially in alkaline environments, inducing a genotoxic effect, which is enhanced by the amount of bioactive molecules present [89,90,91].

To investigate the anti-inflammatory effect of the combined bioactive molecules, macrophages were generated from PMA-induced THP-1 monocytes. Macrophages were exposed to the NLRP3 activation protocol, and the modulation of NLRP3 by the combined bioactive molecules was evaluated. PMA is a molecule capable of inducing the differentiation of monocytes into macrophages through the modulation of some microRNAs, such as mir-155, mir-222, mir-424, and mir-503, which are responsible for controlling the differentiation process of the myeloid lineage [48,92,93,94]. THP-1 monocytes exposed to PMA adhered to the bottom of the cell culture flask and showed changes in terms of morphological characteristics in relation to monocytes not exposed to PMA. Then, THP1-derived macrophages were induced to NLRP3 inflammasome activation by LPS + nigericin exposure.

As expected, MCC950 was able to partially recover cellular conditions compared to the LPS + nigericin activation control. Furthermore, all the combinations of bioactive molecules were able to partially reverse the inflammatory activation in macrophages caused by LPS + nigericin via decreasing the levels of ROS, NO, and dsDNA release. Based on all the assays performed, the combination of catechin + epicatechin was chosen to conduct lactate level determination and gene expression assays. Lactate is a metabolite produced mainly during anaerobic glycolysis, when cells convert glucose into energy in the absence or low presence of oxygen [95]. The relationship between lactate and inflammation is complex, involving several aspects of cellular metabolism and the immune response. During inflammation, immune system cells, such as neutrophils and macrophages, increase their glycolytic metabolism, resulting in greater lactate production [96]. In the tricarboxylic acid (TCA) cycle, the enzyme lactate dehydrogenase B (LDHB) converts lactate to pyruvate, which is further oxidized in the mitochondria and subsequently metabolized by the TCA cycle to citrate, malate, and α-ketoglutarate. These metabolites generated by lactate increase the inflammatory response, with notable examples being citrate, which can induce the production of NO, prostaglandins, and ROS [97]. Furthermore, the increase in and accumulation of lactate can influence the inflammatory environment by regulating local pH, causing acidosis in the extracellular environment, which is an important factor in inflammatory diseases [98].

In contrast, lactate accumulation initiates a process called lactylation, which triggers the polarization of M2-like macrophages in a time-dependent manner, referred to as the “lactate clock”, thus regulating the inflammatory response. In the late phase of inflammation, lactylation induces the transformation of M1-like macrophages into M2-like macrophages through epigenetic mechanisms, thus helping to repair tissue damage caused by inflammation [97]. Yang et al. [99] found that the lactate/GPR81 pathway, a lactate receptor, attenuates the activation of NF-κB and reduces the production of pro-inflammatory cytokines TNF-α and IL-6 in macrophages induced by LPS. Furthermore, this pathway inhibits the activation of the NLRP3 inflammasome, mitigating the inflammatory response and minimizing tissue damage associated with inflammation. Additionally, Zhou et al. [100] also observed that lactate inhibits the TLR/NF-κB signaling pathway and the production of pro-inflammatory factors, promoting the polarization of macrophages in a model of intestinal inflammation. These results further illustrate the lactate dosage in this study and may be an explanation for the decrease in lactate levels in the control of LPS + nigericin activation of the NLRP3 inflammasome. Therefore, lactate plays a fundamental role in inflammatory diseases and represents a potential therapeutic target for their treatment.

MCC950, catechin, epicatechin, and the combination of catechin + epicatechin reduced caspase-1 gene expression in macrophages activated by LPS + nigericin. White et al. [101] showed that flavonoids, such as catechins, act as competitive inhibitors of caspase-1, -3, and -7, suggesting that such flavonoids can have a therapeutic purpose as specific inhibitors of caspases. This is an important finding, since pro-caspase-1 and caspase-1 activated after the assembly of the NLRP3 protein complex are part of the activation of the inflammatory cascade. Additionally, isolated MCC950, catechin, and epicatechin significantly reduced the gene expression of the NLRP3 in cells activated by LPS + nigericin. On the other hand, the catechin + epicatechin combination caused a significant increase in NLRP3 gene expression. It is already known that catechins in general are great antioxidant molecules, with this perhaps being the reason for why these molecules could present an anti-inflammatory effect. However, there are some conditions where catechins could have pro-oxidant activity via autoxidation or peroxidase-based oxidation, as well through the capacity of generating ROS, mainly phenoxyl radicals [102,103]. Additionally, Caro et al. [104] demonstrated that catechin could generate oxidants in a CYP2E1 (a member of the cytochrome P450 family)-dependent way. This result suggests that isolated bioactive molecules modulate the NLRP3 gene more effectively than combined molecules. We believe that by the time we combine catechin and epicatechin (a catechin stereoisomer), we have promoted a condition of too many antioxidants, which allows the molecules to work as pro-oxidants reflexing on NLRP3 gene expression upregulation. Following the inflammatory cascade, isolated MCC950, catechin, and epicatechin were found to also decrease the gene expression of IL-1β, IL-6, and TNF-α, compared to the activated cells. These results were expected since these bioactive molecules were able to reduce the main elements of the NLRP3 inflammasome. IL-10 was not modulated by the treatments performed.

As described, bioactive molecules have significant potential in therapeutic applications; however, the clinical application of flavonoids, in particular, is limited due to chemical instability due to the presence of free hydroxyl groups [105]; low bioavailability, as polyphenols from the diet are predominantly presented in glycosylated form with one or more sugar residues conjugated to a hydroxyl group and/or aromatic ring, representing the main reason for low intestinal absorption, with only a small amount of absorption occurring (5 to 10%) mainly in the colon [23,106]; sensitivity to environmental and biological enzymatic degradation, mainly due to the effects of stomach acids [107]; solubility in water is hampered due to the amphiphilic chemical structure, which suggests affinity for low-polarity environments, due to the presence of a hydrophobic aromatic moiety, and high polarity, due to the presence of hydrophilic hydroxyl groups [108]; and photoinduced oxidation, hindering the use of such molecules as nutraceuticals [109].

These limitations could be mitigated or solved using drug delivery approaches through the encapsulation of flavonoids in formulations that can improve the solubility, stability, and bioavailability of these molecules [109,110]. Therefore, nanotechnology, for example, is a tool capable of protecting from degradation and enhancing the biological effects of flavonoids [111]. Therefore, our results not only support other theoretical findings but strongly suggest the effectiveness of natural compounds derived from açaí, especially epicatechin, both in silico and in vitro in acting as effective modulators of the inflammasome.

## 5. Conclusions

The results indicate that the key molecules in the açaí extract’s chemical matrix, identified as having the highest binding affinity with the NLRP3 PYD domain through in silico methods, are catechin, apigenin, and epicatechin. These molecules, both isolated and combined, exhibited a satisfactory in vitro safety profile. Additionally, catechin, apigenin, and epicatechin showed promising anti-inflammatory potential through the modulation of NLRP3, both individually and in combination. However, while the catechin and epicatechin combination inhibits caspase-1, isolated molecules appear to modulate the NLRP3 inflammasome more effectively than when combined. Consequently, these bioactive molecules, particularly flavonoids, hold potential therapeutic value as anti-inflammatory agents through NLRP3 modulation. The application of technologies like nanotechnology could potentially enhance their biological effects.

## Figures and Tables

**Figure 1 biology-13-00729-f001:**
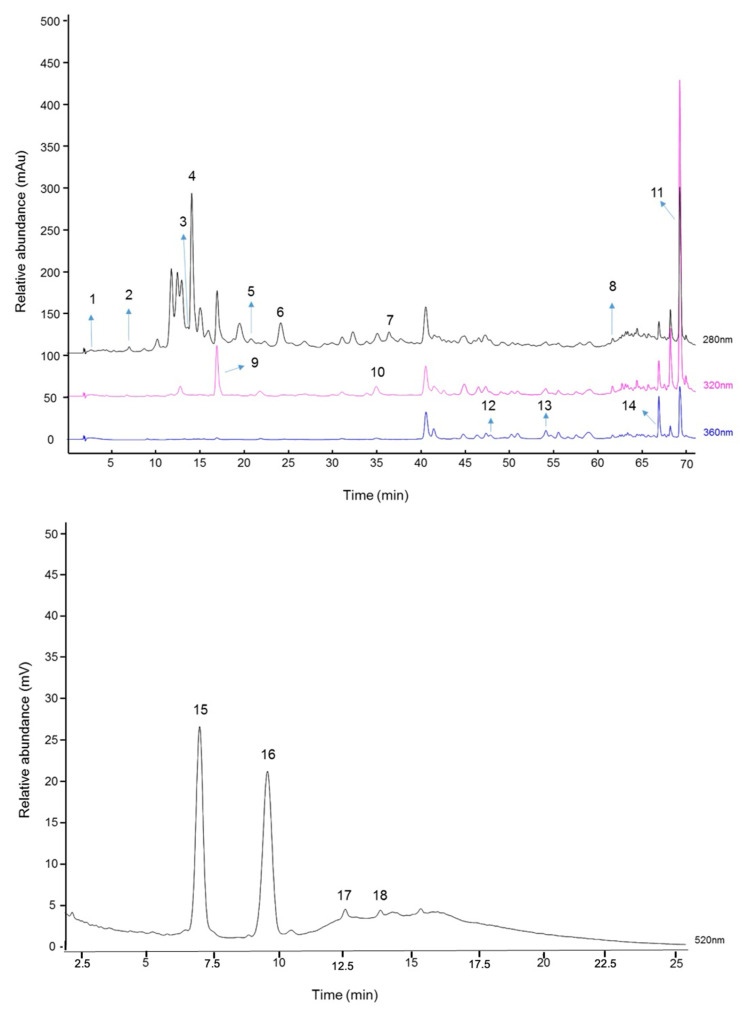
Chromatograms acquired at 280 nm, 320 nm, and 360 nm for non-anthocyanin phenolic compounds and 520 nm for anthocyanin phenolic compounds. Peak 1: gallic acid; peak 2: protocatechuic acid; peak 3: epigallocatechin; peak 4: catechin; peak 5: syringic acid; peak 6: epicatechin; peak 7: taxifolin; peak 8: t-cinnamic acid; peak 9: caffeic acid; peak 10: t-ferulic acid; peak 11: apigenin; peak 12: orienthin; peak 13: kaempferol 3-β-D-glucopyranoside; peak 14: luteolin; peak 15: cyanidin-3-O-glucoside; peak 16: cyanidin-3-O-rutinoside; peak 17: peonidin-3-O-glucoside; peak 18: peonidin-3-O-rutinoside.

**Figure 2 biology-13-00729-f002:**
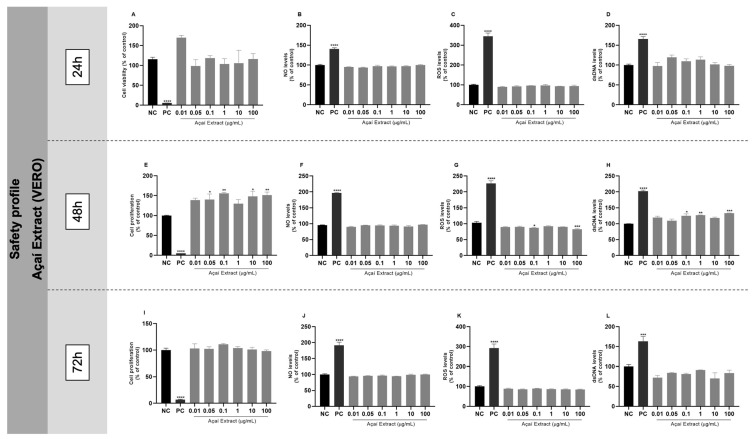
Açaí extract concentration–response curve—in vitro safety profile evaluation. VERO cells were exposed to different concentrations of free açaí extract for 24, 48, and 72 h of incubation. (**A**,**E**,**I**) Assessment of cellular viability (24 h) and proliferation (48 and 72 h) indexes by MTT assay; (**B**,**F**,**J**) measurement of NO levels after 24, 48, and 72 h of incubation, respectively; (**C**,**G**,**K**) measurement of ROS levels after 24, 48, and 72 h of incubation, respectively; (**D**,**H**,**L**) quantification of dsDNA extracellular indexes after 24, 48, and 72 h of incubation, respectively; NC: negative control (cells under conventional cell culture condition); PC: cells exposed to 200 µM of H_2_O_2_ for MTT, DCFH-DA, and PicoGreen assays and 10 µM of sodium nitroprusside for NO determination assay; statistical analysis was performed by one-way ANOVA followed by Tukey post hoc. Results with *p* < 0.05 were considered significant. * *p* < 0.05; ** *p* < 0.01; *** *p* < 0.001; **** *p* < 0.0001.

**Figure 3 biology-13-00729-f003:**
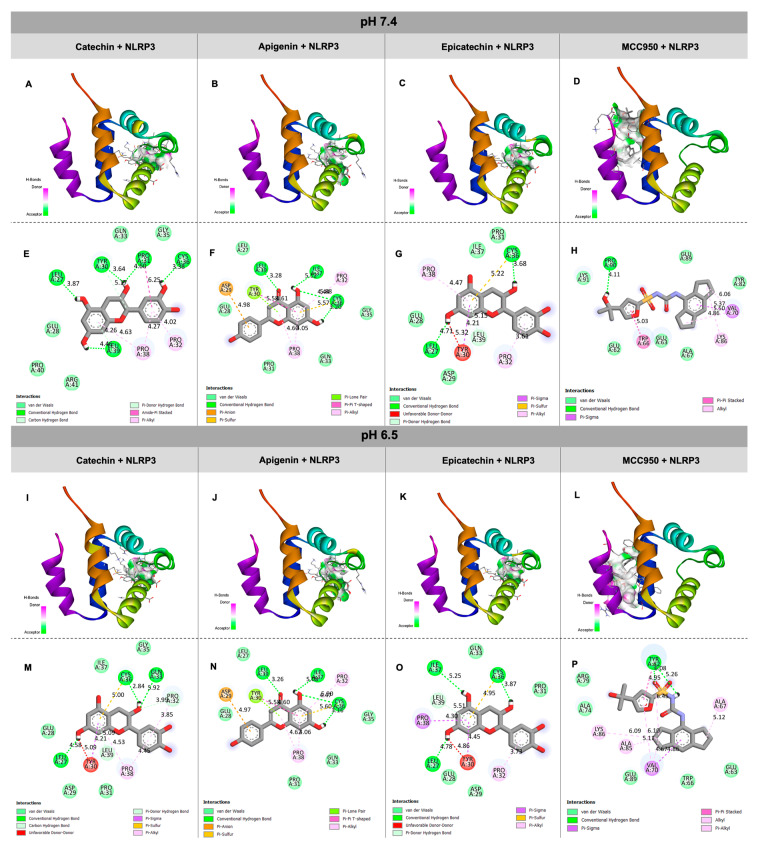
Interactions between NLRP3 (PYD domain) and catechin, apigenin, epicatechin, and MCC950 with pH 7.4 and 6.5. (**A**–**D**,**I**–**L**) Interactions between catechin, apigenin, epicatechin, and MCC950, respectively, with NLRP3 PYD domain; (**E**–**H**,**M**–**P**) 2D map of the interaction of NLRP3 with catechin, apigenin, epicatechin, and MCC950 and amino acid residues and bond types.

**Figure 4 biology-13-00729-f004:**
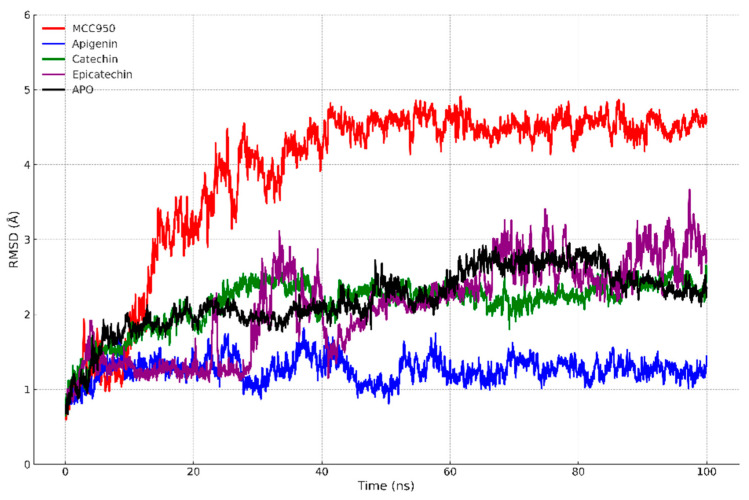
RMSD of the complexes formed between the PYD domain of NLRP3 and the ligands MCC950, apigenin, catechin, epicatechin, and the APO (unbound) form of the protein (without ligand) over 100 ns of molecular dynamics simulation. It is observed that the complex with MCC950 presented greater structural variation, with the RMSD reaching approximately 5 Å, indicating greater conformational flexibility. Apigenin presented the lowest RMSD, around 1.26 Å, suggesting greater stabilization of the protein. Catechin and epicatechin exhibited intermediate RMSD, ranging from 2.10 to 2.16 Å, while the APO form presented moderate fluctuations, with RMSD around 2.20 Å.

**Figure 5 biology-13-00729-f005:**
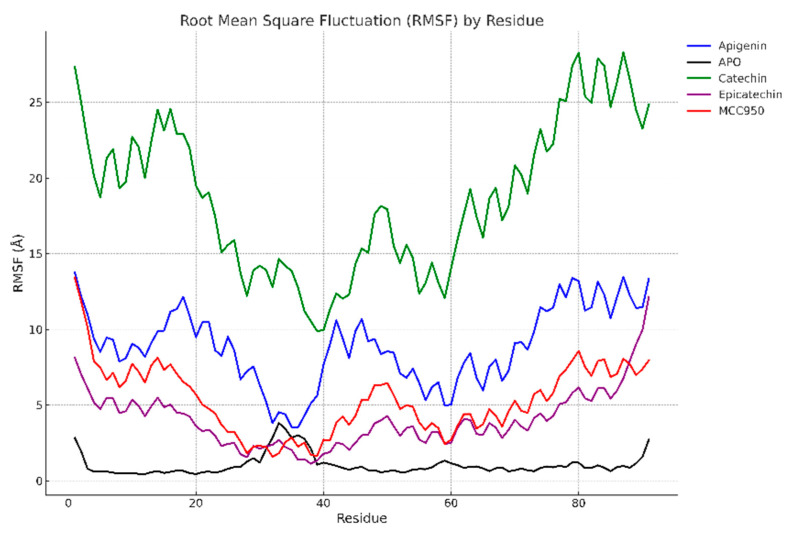
Root Mean Square Fluctuation (RMSF) of the complexes formed between the PYD domain of NLRP3 and the ligands MCC950, apigenin, catechin, epicatechin, and the APO form of the protein (without ligand). The fluctuation was calculated over 100 ns of molecular dynamics simulation. Catechin showed the highest fluctuations in residues, particularly in the terminal and central regions, suggesting greater conformational flexibility. Apigenin exhibited moderate fluctuations, while epicatechin and MCC950 displayed lower residual fluctuation, suggesting more efficient conformational stabilization in these critical regions. The APO form exhibited the lowest variation, reflecting the absence of interactions with ligands. Stabilization in the regions between Leu20 and Pro40 was observed in the MCC950 and epicatechin complexes, which may indicate more stable interactions in these key residues.

**Figure 6 biology-13-00729-f006:**
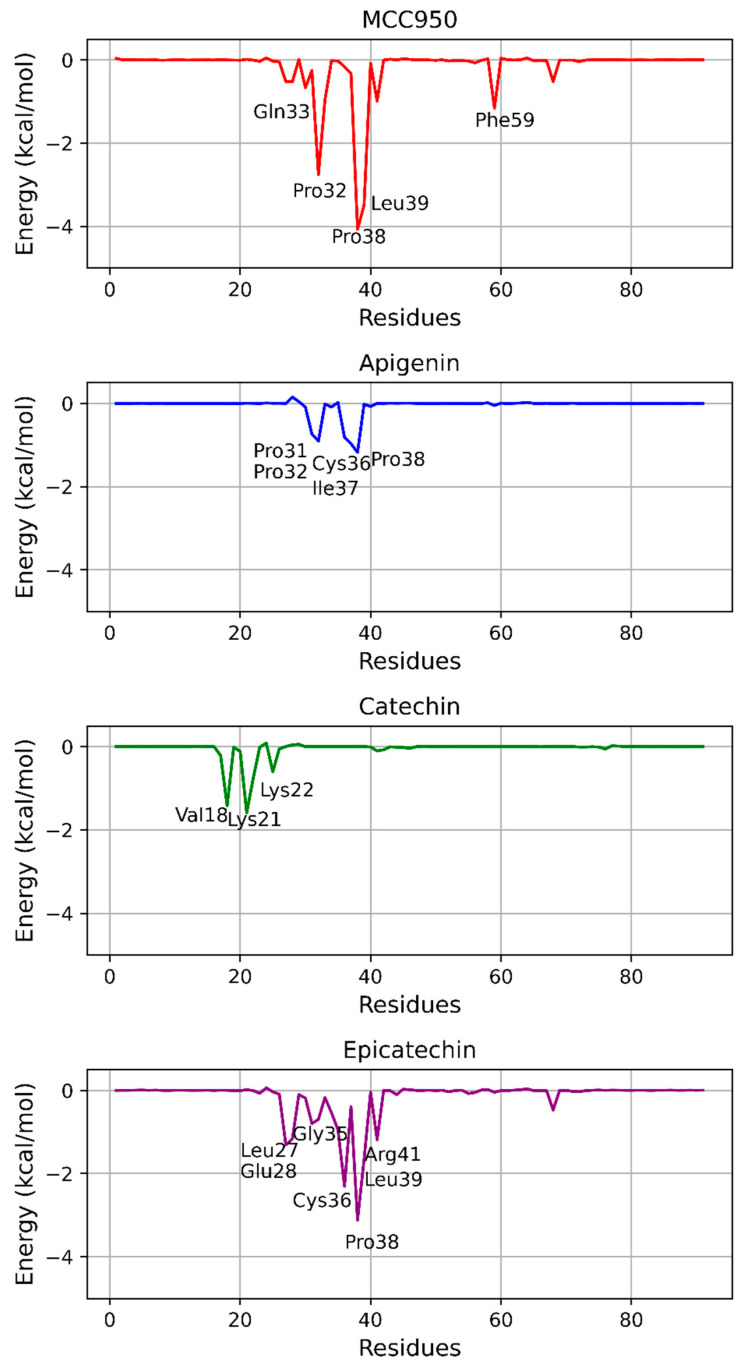
Energy decomposition per residue for the complexes formed between the PYD domain of NLRP3 and the ligands MCC950, apigenin, catechin, and epicatechin. The energetic contribution of each residue is presented in kcal/mol, highlighting the most critical residues for the stabilization of the complexes. For MCC950, residues Gln33, Pro32, Pro38, and Leu39 showed significant contributions, with energy values below −2 kcal/mol, indicating strong interactions at these sites. For apigenin, residues Pro31, Pro32, Cys36, and Pro38 were the main energetic contributors, while the catechin complexes presented more relevant interactions at residues Lys21, Lys22, and Val18. Lastly, for epicatechin, residues Cys36, Pro38, Leu39, and Arg41 were highlighted as the main stabilizers.

**Figure 7 biology-13-00729-f007:**
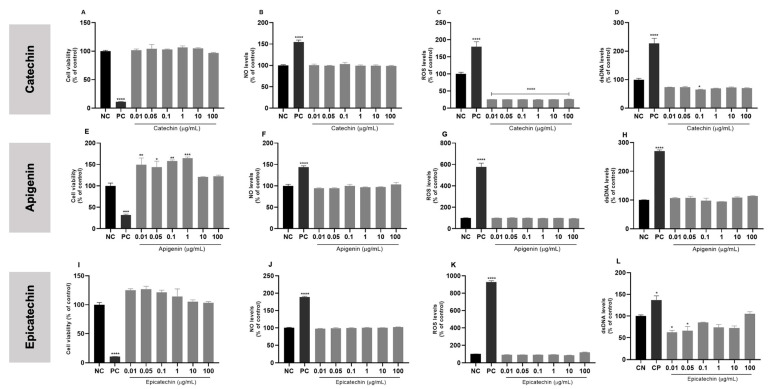
Bioactive molecule concentration curve—in vitro safety profile evaluation. VERO cells were exposed to different concentrations of catechin, apigenin, and epicatechin for 24 h of incubation. (**A**,**E**,**I**) Assessment of cellular viability (24 h) indexes by MTT assay; (**B**,**F**,**J**) measurement of NO levels after 24 h of incubation; (**C**,**G**,**K**) measurement of ROS levels after 24 h; (**D**,**H**,**L**) quantification of dsDNA extracellular indexes after 24 h of incubation; NC: negative control (cells under conventional cell culture condition); PC: cells exposed to 200 µM of H_2_O_2_ for MTT, DCFH-DA, and PicoGreen assays and 10 µM of sodium nitroprusside for NO determination assay; statistical analysis was performed by one-way ANOVA followed by Tukey post hoc. Results with *p* < 0.05 were considered significant. * *p* < 0.05; ** *p* < 0.01; *** *p* < 0.001; **** *p* < 0.0001.

**Figure 8 biology-13-00729-f008:**
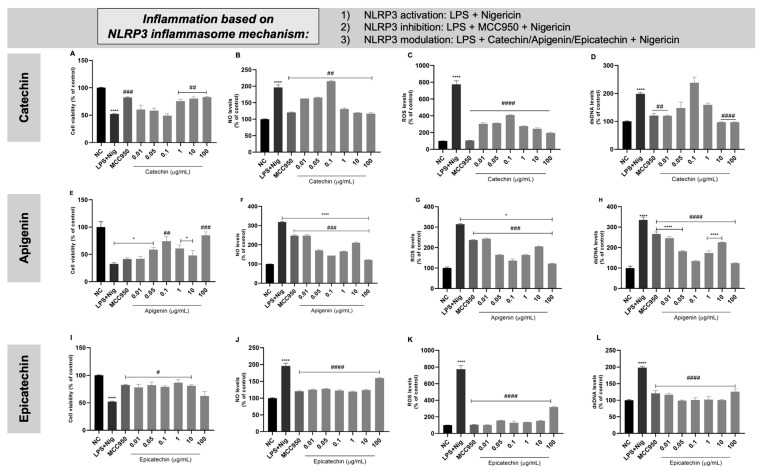
Anti-inflammatory capacity of catechin, apigenin, and epicatechin in monocytes. LPS + nigericin was used as the NLRP3 activation agent; MCC950 was used as a known NLRP3 inhibitor agent. (**A**,**E**,**I**) Assessment of cellular viability indexes by MTT assay; (**B**,**F**,**J**) indirect determination of NO levels; (**C**,**G**,**K**) qualitative measurement of ROS production; (**D**,**H**,**L**) quantification of dsDNA extracellular indexes. NC: negative control (cells under conventional cell culture condition); statistical analysis was performed by one-way ANOVA followed by Tukey post hoc. Results with *p* < 0.05 were considered significant. * represents comparison to the negative control; # represents comparison to LPS positive control; * *p* < 0.05; **** *p* < 0.0001; ## < 0.01; ### < 0.001; #### < 0.0001.

**Figure 9 biology-13-00729-f009:**
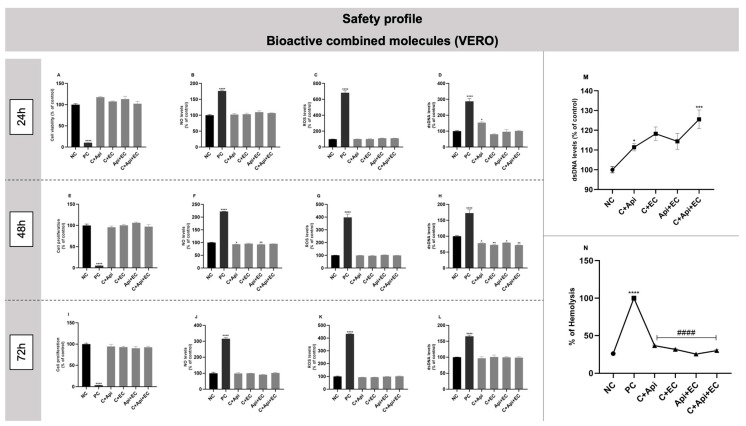
Combined bioactive molecules—in vitro safety profile evaluation. VERO cells were exposed to different combinations of bioactive molecules for 24, 48, and 72 h of incubation. (**A**,**E**,**I**) Assessment of cellular viability (24 h) and proliferation (48 and 72 h) indexes by MTT assay; (**B**,**F**,**J**) measurement of NO levels after 24, 48, and 72 h of incubation, respectively; (**C**,**G**,**K**) measurement of ROS levels after 24, 48, and 72 h of incubation, respectively; (**D**,**H**,**L**) quantification of dsDNA extracellular indexes after 24, 48, and 72 h of incubation, respectively—NC: negative control (cells under conventional cell culture condition); PC: cells exposed to 200 µM of H_2_O_2_ for MTT, DCFH-DA, and PicoGreen assays and 10 µM of sodium nitroprusside for NO determination assay; (**M**) assessment of genotoxic effect; (**N**) measurement of hemolysis—PC: red blood cells were lysed with dH_2_O. C: catechin; Api: apigenin; and EC: epicatechin. Statistical analysis was performed by one-way ANOVA followed by Tukey post hoc. # represents comparison to hemolysis positive control. Results with *p* < 0.05 were considered significant. * *p* < 0.05; ** *p* < 0.01; *** *p* < 0.001; **** *p* < 0.0001; #### < 0.0001.

**Figure 10 biology-13-00729-f010:**
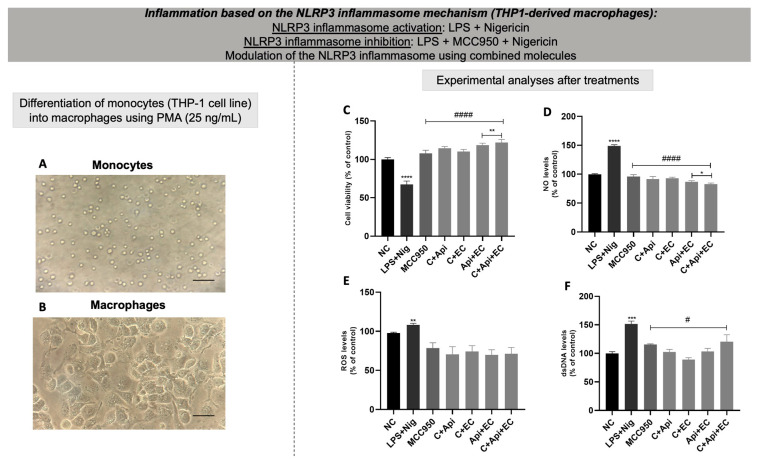
Inflammation based on the NLRP3 inflammasome mechanism. LPS + nigericin was used as the activation agent; MCC950 was used as a known inhibitor agent. Left side—THP-1 differentiation to macrophages: (**A**) microscopical analysis of THP1 monocytes without any treatment; (**B**) THP1-derived macrophages generated by PMA exposure. Right side—experimental analyses of (**C**) cell viability by MTT; (**D**) NO indexes; (**E**) ROS levels; and (**F**) measurement of the extracellular dsDNA index. NC: negative control (cells under conventional cell culture condition). C: catechin; Api: apigenin; and EC: epicatechin. Statistical analysis was performed by one-way ANOVA followed by Tukey post hoc. Results with *p* < 0.05 were considered significant. * represents comparison to the negative control; # represents comparison to LPS positive control; * *p* < 0.05; ** *p* < 0.01; *** *p* < 0.001; **** *p* < 0.0001; #### < 0.0001. Magnification: 20×.

**Figure 11 biology-13-00729-f011:**
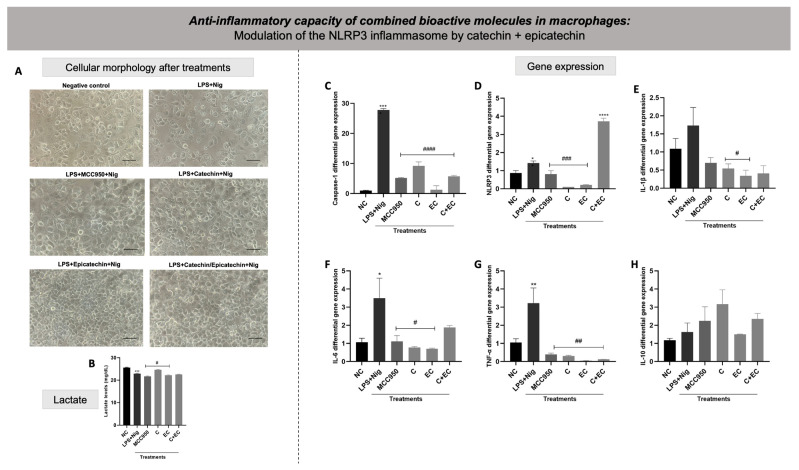
Anti-inflammatory capacity of combined bioactive molecules in macrophages. (**A**) Analysis of cell morphology by optical microscopy after each treatment. (**B**) Lactate levels after each treatment. (**C**) Caspase-1 gene expression after each treatment. (**D**) NLRP3 gene expression after each treatment. (**E**) IL-1β gene expression after each treatment. (**F**) IL-6 gene expression after each treatment. (**G**) TNF-α gene expression after each treatment. (**H**) IL-10 gene expression after each treatment. NC: negative control (cells under conventional cell culture condition). C: catechin; EC: epicatechin. Statistical analysis was performed by one-way ANOVA followed by Tukey post hoc. Results with *p* < 0.05 were considered significant. * represents comparison to the negative control; # represents comparison to LPS positive control; * *p* < 0.05; ** *p* < 0.01; *** *p* < 0.001; **** *p* < 0.0001; ## < 0.01; ### < 0.001; #### < 0.0001. Magnification: 20×.

**Table 1 biology-13-00729-t001:** Molecular docking results of ligand interactions against the NLRP3 PYD domain.

Target	Ligand	Affinity (kcal/mol)		RMSD (Å)	
		pH 7.4	pH 6.5	pH 7.4	pH 6.5
NLRP3 PYD	Catechin	−7.52	−7.0	1.126	1.808
	Apigenin	−7.1	−7.1	0.809	0.945
	Epicatechin	−6.3	−6.3	1.740	1.586
	Taxifolin	−6.2	−6.1	0.663	0.783
	Epigallocatechin	−6.2	−6.2	0.828	0.919
	MCC950	−7.5	−7.4	1.977	1.934

**Table 2 biology-13-00729-t002:** RMSD, SASA, and Gyration Radius (Rg) averages and corresponding standard deviations for the complexes formed between the PYD domain of NLRP3 and the ligands MCC950, apigenin, catechin, epicatechin, and the APO form.

	RMSD		SASA		Rg	
System	Average (Å)	Standard Deviation (Å)	Average (Å2)	Standard Deviation (Å2)	Average (Å)	Standard Deviation (Å)
APO	2.20	0.41	1078.85	69.62	11.79	0.16
Apigenin	1.26	0.17	1027.80	51.02	11.79	0.09
Catechin	2.16	0.33	1030.49	67.84	11.88	0.09
Epicatechin	2.10	0.63	1093.74	59.10	11.98	0.15
MCC950	3.90	1.08	1155.29	42.17	11.97	0.21

**Table 3 biology-13-00729-t003:** Binding free energy components calculated using the MM/GBSA method for the complexes formed between the PYD domain of NLRP3 and the ligands.

Molecule	∆E_vdW_	∆E_ele_	∆G_GB_	∆G_nonpol_	∆G_MM/GBSA_
Apigenin	−12.0498 ± 0.1329	−4.2016 ± 0.1591	10.6960 ± 0.1818	−1.5467 ± 0.0154	−7.1022 ± 0.0951
Catechin	−12.1843 ± 0.1093	−8.7908 ± 0.2185	16.5944 ± 0.2245	−1.9396 ± 0.0163	−6.3203 ± 0.0801
Epicatechin	−32.2019 ± 0.0581	−17.7627 ± 0.1384	32.8777 ± 0.1192	−4.0496 ± 0.0049	−21.1365 ± 0.0733
MCC950	−41.4292 ± 0.0787	−15.9719 ± 0.1908	31.2462 ± 0.1930	−4.5383 ± 0.0111	−30.6931 ± 0.0744

## Data Availability

Data will be available on request.

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
