# Peer review of "Euterpe oleracea Mart. Bioactive Molecules: Promising Agents to Modulate the NLRP3 Inflammasome"

_biology, 2024, doi:10.3390/biology13090729_

Round 1
Reviewer 1 Report
Comments and Suggestions for Authors
After reviewing the manuscript, I found it interesting. However, I think there are some points that need clarification.
In the in silico part, the authors employed docking simulation to study the affinity between the ligands and the protein receptor. Generally, depending only on docking simulations without additional methods to study protein dynamics (e.g., MD simulations) could lead to false-positive results. Therefore, I recommend that the authors perform MD simulations followed by MM-PBSA or MM-GBSA calculations to investigate the binding affinity in more depth. It is also essential that any computational methods should be validated to ensure reliability. If performing MD simulations is not possible, I suggest that the authors either exclude the docking section from the manuscript or highly reduce its emphasis (perhaps moving it to the supplementary).
Here is my minor comments:
1- Generally, NLRP3 is a secondary pathway for TLRs, particularly TLR4 and TLR2. Additionally, the NOD-like receptors (NOD1 and NOD2) can regulate NLRP3 in immunity and homeostasis. By downregulating NOD, NLRP3 can also be regulated. I recommend that the authors consider investigating NOD expression in their experiments.
2- To study inflammation through the TLR and NLRP3 signaling pathways, it is important to check the gene expression of pro-inflammatory cytokines (e.g., TNFA, IL1B, IL8) as well as anti-inflammatory cytokines such as TGFB and IL10.
3- In Figure 8, the upregulation of NLRP3 gene expression by catechin+epicatechin is unexpected, as downregulation would have been more anticipated. Could the authors clarify this finding?
Author Response
September 2024
To: Reviewer
Dear Reviewer,
Thank you for considering our manuscript entitled "Euterpe oleracea Mart. Bioactive Molecules: promising agents to modulate the NLRP3 inflammasome" for publication in Biology (Section: Biology and Function of Inflammasomes). I am sending to you the revised version of our manuscript.
All the reviewers’ comments were carefully considered, and the appropriate changes were applied. We highlighted the changes in red.
We would greatly appreciate an opportunity to publish our manuscript in Biology. Thank you for your time and consideration. Please do not hesitate to let us know if you have any concerns or questions.
Yours Sincerely,
Alencar Kolinski Machado
Ana Cristina Andreazza
Corresponding authors
Reviewer comments:
General comment: “After reviewing the manuscript, I found it interesting. However, I think there are some points that need clarification.”
- In the in silico part, the authors employed docking simulation to study the affinity between the ligands and the protein receptor. Generally, depending only on docking simulations without additional methods to study protein dynamics (e.g., MD simulations) could lead to false-positive results. Therefore, I recommend that the authors perform MD simulations followed by MM-PBSA or MM-GBSA calculations to investigate the binding affinity in more depth. It is also essential that any computational methods should be validated to ensure reliability. If performing MD simulations is not possible, I suggest that the authors either exclude the docking section from the manuscript or highly reduce its emphasis (perhaps moving it to the supplementary).
Response: Thank you for your comment. We have performed the molecular dynamics simulations during 100 ns (nanoseconds) followed by the free energy calculations by MM/GBSA which were estimated for each of the systems complexed with the main molecules found in our natural health product. This analysis was added in the manuscript in the Material and Methods, presenting new results and discussions. Additionally, João Augusto Pereira da Rocha was the researcher who developed this in silico analysis. In this regard, his name was included as a co-author of this article.
General comment: Here is my minor comments:
- Generally, NLRP3 is a secondary pathway for TLRs, particularly TLR4 and TLR2. Additionally, the NOD-like receptors (NOD1 and NOD2) can regulate NLRP3 in immunity and homeostasis. By downregulating NOD, NLRP3 can also be regulated. I recommend that the authors consider investigating NOD expression in their experiments.
Response: Thank you for your suggestion. Unfortunately, we do not have primers to investigate the NOD expression and due to the restrict time to reply the reviewers comments we were unable to purchase it and carry this analysis out. However, for sure we will consider this suggestion for further projects.
- To study inflammation through the TLR and NLRP3 signaling pathways, it is important to check the gene expression of pro-inflammatory cytokines (e.g., TNFA, IL1B, IL8) as well as anti-inflammatory cytokines such as TGFB and IL10.
Response: Thank you for this great suggestion. We have performed the gene expression of interleukin-1beta (IL-1b), interleukin-6 (IL-6), tumor necrosis factor-alpha (TNF-a), and interleukin-10 (IL-10) in THP1-derived macrophages for our experiments. In this regard, figure 8 was changed as well as the new results were described in the Results section as: “Macrophages exposed to LPS+nigericin showed significant higher differential gene expression of caspase-1, NLRP3, IL-6, and TNF-α compared to untreated cells (figures 8C, 8D, 8F, and 8G, respectively). Cells exposed to LPS+nigericin also present higher levels of IL-1β, however, this result was not significant (figure 8E). Regarding the IL-10 (figure 8H), macrophages exposed to LPS+nigericin did not present significant changes compared to the negative control. Macrophages activated by LPS+nigericin and treated with MCC950, catechin and epicatechin isolated showed reduced gene expression of caspase-1, NLRP3, IL-6, and TNF-α compared to the LPS+nigericin positive control. Despite LPS+nigericin exposure not increasing IL-1β levels, it was observed that catechin and epicatechin were able to reduce the gene expression of this cytokine in comparison to the positive control. On the other hand, treatment with the combined molecules was able to reduce the gene expression of caspase-1 and TNF-α, but not NLRP3, IL-1β or IL-6. A significant increase in the expression of the NLPR3 was observed under catechin+epicatechin treatment.” The information collected from these results has been added into the Discussion section.
- In Figure 8, the upregulation of NLRP3 gene expression by catechin+epicatechin is unexpected, as downregulation would have been more anticipated. Could the authors clarify this finding?
Response: Thank you for pointing this out. The NLRP3 gene expression upregulation observed under catechin+epicatechin was unexpected. It is already known that catechins in general are great antioxidant molecules, being able to inhibit the oxidative activity of several different molecules avoiding or attenuating the oxidative stress. This is also the reason why catechins could potentially act as anti-inflammatory agents since reactive oxygen species (ROS) may act as damage-associated molecular patterns (DAMP). However, in some conditions these bioactive molecules could perform a pro-oxidative activity. There are some evidence showing that the pro-oxidant mechanisms of catechins may be related to autoxidation or peroxidase-based oxidation, as well capacity of generating ROS, mainly phenoxyl radicals (Nakayama et al., 1995; Mochizuki et al., 2002). Additionally, Caro et al (2019) demonstrated that catechin could generate oxidants in a CYP2E1 (a member of the cytochrome P450 family)-dependent way. In this regard, we believe that by the time we combine catechin and epicatechin (a catechin stereoisomer) we have promoted a condition of too many antioxidants, which allowed the molecules to work as pro-oxidant reflexing on a NLRP3 gene expression upregulation. This explanation was added to the article as: “It is already known that catechins in general are great antioxidant molecules, perhaps being this the reason of why these molecules could present anti-inflammatory effect. However, there are some conditions where catechins could have pro-oxidant activity via autoxidation or peroxidase-based oxidation, as well through the capacity of generating ROS, mainly phenoxyl radicals [1,2]. Additionally, Caro et al [3] demonstrated that catechin could generate oxidants in a CYP2E1 (a member of the cytochrome P450 family)-dependent way. This result suggests that isolated bioactive molecules modulate the NLRP3 gene more effectively than combined. We believe that by the time we combine catechin and epicatechin (a catechin stereoisomer) we have promoted a condition of too many antioxidants, which allowed the molecules to work as pro-oxidant reflexing on a NLRP3 gene expression upregulation.”
Reviewer 2 Report
Comments and Suggestions for Authors
That is a nice working hypothesis. However, there are some points in the paper that should be checked.
1) Paragraph 2.3 is 90% identical to paragraph 2.5.1
2) lines 191-204 - different front
3) Detection of NO by Griss- is an indirect method for determining NO (nitrite and nitrate), and SNP as a NO donor has a short life. The authors may think about a future detection of NO using another method (fluorimetric?) and a confirmation of the modulation of NO release by detecting the expression levels of the iNOS protein.
4) line 273 - When you write a protocol for a scientific journal, you use the centrifugal force expressed as a multiple of g, the symbol
for normal gravitational force; otherwise, if I want to reproduce your protocol in my laboratory, I can't because I don't know what centrifuge you have, what radius the centrifuge rotor has, etc.
5) line-337-348-front
6) line-338-348- I'm confused about viability at 24h, proliferation at 48h and 72h (?), and the correlation with dsDNA.
7) line-461-462- try to frame the figure so that the phrase has fluidity
8) check the article for in vitro vs in vitro
9) also check that once an abbreviation has been made, it must be kept throughout the article. It isn't easy to follow the re-abbreviation of terms.
10) Please give an explanation why the lactate levels test was chosen
11) 1) Please use your full name as the author. I found authors who lost their middle names in the article's references. See also author names vs author contributions.
Author Response
September 2024
To: Reviewer
Dear Reviewer,
Thank you for considering our manuscript entitled "Euterpe oleracea Mart. Bioactive Molecules: promising agents to modulate the NLRP3 inflammasome" for publication in Biology (Section: Biology and Function of Inflammasomes). I am sending to you the revised version of our manuscript.
All the reviewers’ comments were carefully considered, and the appropriate changes were applied. We highlighted the changes in red.
We would greatly appreciate an opportunity to publish our manuscript in Biology. Thank you for your time and consideration. Please do not hesitate to let us know if you have any concerns or questions.
Yours Sincerely,
Alencar Kolinski Machado
Ana Cristina Andreazza
Corresponding authors
Reviewer comments:
General comment: “That is a nice working hypothesis. However, there are some points in the paper that should be checked.”
1) Paragraph 2.3 is 90% identical to paragraph 2.5.1
Response: Thank you for pointing it out. The 2.5.1 paragraph has been changed to: “VERO cells were cultivated, maintained, and tested as described in item 2.3. However, in this case, the performed treatments were conducted with catechin, apigenin, or epicatechin (0.01-100 µg/mL).”
2) lines 191-204 - different front
Response: This mistake was addressed.
3) Detection of NO by Griss- is an indirect method for determining NO (nitrite and nitrate), and SNP as a NO donor has a short life. The authors may think about a future detection of NO using another method (fluorimetric?) and a confirmation of the modulation of NO release by detecting the expression levels of the iNOS protein.
Response: Thank you for this valuable suggestion. Detecting the indirect levels of nitric oxide using Greiss reagent has been very well stablished and published in several studies. However, we agree that there are other methods to measure it. As a research group we will plan to implement new techniques to analyze nitric oxide levels for further investigations.
4) line 273 - When you write a protocol for a scientific journal, you use the centrifugal force expressed as a multiple of g, the symbol for normal gravitational force; otherwise, if I want to reproduce your protocol in my laboratory, I can't because I don't know what centrifuge you have, what radius the centrifuge rotor has, etc.
Response: Thank you for pointing this out. All the centrifuge information that was in RPM was changed to “x g”.
5) line-337-348-front
Response: This mistake was addressed.
6) line-338-348- I'm confused about viability at 24h, proliferation at 48h and 72h (?), and the correlation with dsDNA.
Response: According to the THP-1 cell line data sheet file, these cells could duplicate in about 2-3 days. In this regard, we considered that the evaluation performed with MTT after 24h of incubation reflets the direct number of viable cells, while the analysis developed after 48 and 72h are related to the cellular index of proliferation. We have measured extracellular dsDNA release in the cells’ supernatant. When cells suffer membranes damages, they release the dsDNA to the extracellular environment, allowing the determination of this molecule as a marker of cellular mortality. This is the reason why the dsDNA measurement was correlated to the MTT results. To clarify it, the following sentences were added in the Methods: “Cell viability (24h of incubation) and proliferation (48 and 72h of incubation – considering the cellular duplication rate) were evaluated using the 3-(4,5-dimethylthiazol-2-yl)-2,5-diphenyltetrazolium (MTT) bromide assay...” and “Considering that when cells suffer membrane damages the dsDNA is released to the extracellular environment, this measurement could reflect an index of cellular mortality.”
7) line-461-462- try to frame the figure so that the phrase has fluidity
Response: This figure has been framed correctly.
8) check the article for in vitro vs in vitro
Response: The revision was made.
9) also check that once an abbreviation has been made, it must be kept throughout the article. It isn't easy to follow the re-abbreviation of terms.
Response: All the used abbreviation were revised.
10) Please give an explanation why the lactate levels test was chosen
Response: To explain why we decided to measure lactate levels it was added the following information in the Discussion section: Lactate is a metabolite produced mainly during anaerobic glycolysis, when cells convert glucose into energy in the absence or low presence of oxygen [4]. The relationship between lactate and inflammation is complex, involving several aspects of cellular metabolism and the immune response. During inflammation, immune system cells, such as neutrophils and macrophages, increase their glycolytic metabolism, resulting in greater lactate production [5]. In the tricarboxylic acid (TCA) cycle, the enzyme lactate dehydrogrenase B (LDHB) converts lactate to pyruvate, which is further oxidized in the mitochondria and subsequently metabolized by the TCA cycle to citrate, malate, and α-ketoglutarate. These metabolites generated by lactate increase the inflammatory response, notable examples being citrate, which can induce the production of NO, prostaglandins, and ROS [6]. Furthermore, the increase and accumulation of lactate can influence the inflammatory environment by regulating local pH, causing acidosis in the extracellular environment, which is an important factor in inflammatory diseases [7].
In contrast, lactate accumulation initiates a process called lactylation, which triggers the polarization of M2-like macrophages in a time-dependent manner, referred to as the “lactate clock,” thus regulating the inflammatory response. In the late phase of inflammation, lactylation induces the transformation of M1-like macrophages into M2-like macrophages through epigenetic mechanisms, thus helping to repair tissue damage caused by inflammation [6]. Yang et al. [8] found that the lactate/GPR81 pathway, a lactate receptor, attenuates the activation of NF-κB and reduces the production of pro-inflammatory cytokines TNF-α and IL-6 in macrophages induced by LPS. Furthermore, this pathway inhibits the activation of the NLRP3 inflammasome, mitigating the inflammatory response and minimizing tissue damage associated with inflammation. Additionally, Zhou et al. [9] also observed that lactate inhibits the TLR/NF-κB signaling pathway and the production of pro-inflammatory factors, promoting the polarization of macrophages in a model of intestinal inflammation. These results corroborate the lactate dosage in this study and may be an explanation for the decrease in lactate levels in the control of LPS+nigericin activation of the NLRP3 inflammasome. Therefore, lactate plays a fundamental role in inflammatory diseases and represents a potential therapeutic target for their treatment.
11) Please use your full name as the author. I found authors who lost their middle names in the article's references. See also author names vs author contributions.
Response: This revision was made.
Reviewer 3 Report
Comments and Suggestions for Authors
Thank you for your efforts. Please consider these comments for improvement:
- The authors wrote "Non-anthocyanin phenolics purified fraction from Euterpe oleracea were analyzed .....". How the authors know that these fractions are free from anthocyanins? and what is the problem associated with the presence of anthocyanins, if any. Please discuss and improve the text in the manuscript.
- The authors wrote "Injection volume was 20 μL". Please indicate of what concentration?
- The quantification data of the compounds should be added. Did the authors establish the method of quantification, confirm the validation analytical criteria of the quantification (accuracy, precision, LOD, LOQ, .... etc).
- Section 3. In 3.1, the authors identified 18 compounds? how the identification process was performed? if it is using MS analysis, if so, what about the level of identification, difference in MS analysis measured, obtained, MS fragmentation, ... etc. Please tabulate all of these, in addition, the quantification process of all of these compounds remains vague. Please add more details.
- The authors selected two pHs to run the computational simulation. What is the reason for such selection ? I can see that the results show almost similar data for those pHs.
- What is the standard ligand used for the docking experiment, it is written MCC950?
- Did the color of compounds affect the bioassays? and how the authors ensure that, specifically, these compounds are colored yellow and the conducted experiments are colorimetric ones.
Comments on the Quality of English LanguageModerate editing of English language required.
Author Response
September 2024
To: Reviewer
Dear Reviewer,
Thank you for considering our manuscript entitled "Euterpe oleracea Mart. Bioactive Molecules: promising agents to modulate the NLRP3 inflammasome" for publication in Biology (Section: Biology and Function of Inflammasomes). I am sending to you the revised version of our manuscript.
All the reviewers’ comments were carefully considered, and the appropriate changes were applied. We highlighted the changes in red.
We would greatly appreciate an opportunity to publish our manuscript in Biology. Thank you for your time and consideration. Please do not hesitate to let us know if you have any concerns or questions.
Yours Sincerely,
Alencar Kolinski Machado
Ana Cristina Andreazza
Corresponding authors
Reviewer comments:
General comment: Thank you for your efforts. Please consider these comments for improvement:
1) The authors wrote "Non-anthocyanin phenolics purified fraction from Euterpe oleracea were analyzed .....". How the authors know that these fractions are free from anthocyanins? and what is the problem associated with the presence of anthocyanins, if any. Please discuss and improve the text in the manuscript.
Response: The protocol for the separation of anthocyanin and non-anthocyanin phenolics used in this study was developed and validated by the research group of Dr. Mônica Giusti (Ohio State University, USA), which is one of the main world references on this subject. Furthermore, during the chromatographic run of the non-anthocyanin fraction, monitoring was carried out at 540 nm and no peaks were observed. We chose to separate anthocyanins from other phenolics because we observed that in samples where the concentration of these compounds is the majority in relation to other compounds, such as in Euterpe oleracea, they cause changes in the baseline of the chromatogram, compromising the quantification of non-anthocyanin phenolics. As our objective was also to quantify, we chose to separate them using SPE.
2) The authors wrote "Injection volume was 20 μL". Please indicate of what concentration?
Response: The concentration of the injected extract solution was approximately 20 mg/mL.
3) The quantification data of the compounds should be added. Did the authors establish the method of quantification, confirm the validation analytical criteria of the quantification (accuracy, precision, LOD, LOQ, .... etc).
Response: Thanks for the excellent comment. Yes, the chromatographic methods used have been validated and are published. Detailed information about repeatability, precision, accuracy, etc. can be consulted in the studies 019) and Silva et al. (2020). We included in the text of the manuscript (section 2.2) that the methods used were validated.
4) Section 3. In 3.1, the authors identified 18 compounds? how the identification process was performed? if it is using MS analysis, if so, what about the level of identification, difference in MS analysis measured, obtained, MS fragmentation, ... etc. Please tabulate all of these, in addition, the quantification process of all of these compounds remains vague. Please add more details.
Response: In this study, phenolic compounds were quantified and identified based on analytical standards with a very high degree of purity. Quantification was performed using calibration curves for each of the compounds reported in the study. Information about the analytical curves used for quantification was included in the manuscript (see section 2.2). Identification was carried out based on the elution order and retention of authentic standards and the spectral data obtained from UV–visible absorption spectra. Furthermore, cyanidin and peonidin derivatives were identified based on the order of elution and absorption spectrum according to Lee (2019) and Alcázar-Alay et al. (2017).
References:
- Lee, Anthocyanins of açai products in the United States, NFS J. 14–15 (2019) 14–21. https://doi.org/10.1016/j.nfs.2019.05.001.
S.C. Alcázar-Alay, F.P. Cardenas-Toro, J.F. Osorio-Tobón, G.F. Barbero, M.A.A. Meireles. Obtaining anthocyanin-rich extracts from frozen açai (Euterpe oleracea Mart.) pulp using pressurized liquid extraction. Food Science and Technology, 37(Suppl. 1): 48-54, 2017. https://doi.org/10.1590/1678-457X.33016
5) The authors selected two pHs to run the computational simulation. What is the reason for such selection ? I can see that the results show almost similar data for those pHs.
Response: To explain why we chose two pHs to conduct molecular docking we have added the following information to the Discussion section: These parameters were used because under normal conditions, physiological pH is tightly maintained and regulated at the cellular, tissue, and systemic levels. However, altered pH, especially acidic pH, is associated with physiological and pathological inflammation, infections, tumors, and other disturbances in body homeostasis [10]. In general, this occurs because during inflammation, immune cells infiltrate the tissue, resulting in increased oxygen and glucose demand, which increases lactic acid secretion and consequently acidification of pH [11]. Additionally, activated macrophages play a central role in the pathophysiology and progression of many chronic inflammatory diseases, since they secrete large amounts of pro-inflammatory cytokines and chemokines, such as IL-1β and IL-18. The maturation and secretion of these cytokines are regulated by the NLRP3 inflammasome pathway which under inflammatory conditions, activates the pro-inflammatory cytokine cascade contributing to the progression and severity of inflammatory events [12].
6) What is the standard ligand used for the docking experiment, it is written MCC950?
Response: The standard ligand used for the molecular docking was the MCC950, a well-known NLRP3 inhibitor widely recognized and validated in previous studies (references cited below). The MCC950 was chosen due to its efficacy and use in other investigations. We have performed a molecular dynamics evaluation with free energy calculations, which corroborates with the molecular docking and the in vitro experiments, confirming the stability and the affinity of the MCC950 complex with the NLRP3, as well as with the other systems simulated.
References:
- V. Swanson, M. Deng, J.P.Y. Ting, The NLRP3 inflammasome: molecular activation and regulation to therapeutics, Nat. Rev. Immunol. 19 (2019) 477–489. https://doi.org/10.1038/s41577-019-0165-0.
- Casali, S.A. Serapian, E. Gianquinto, M. Castelli, M. Bertinaria, F. Spyrakis, G. Colombo, NLRP3 monomer functional dynamics: From the effects of allosteric binding to implications for drug design, Int. J. Biol. Macromol. 246 (2023) 125609. https://doi.org/10.1016/j.ijbiomac.2023.125609.
- Yin, J. Lei, J. Yu, W. Cui, A.L. Satz, Y. Zhou, H. Feng, J. Deng, W. Su, L. Kuai, Assessment of AI-Based Protein Structure Prediction for the NLRP3 Target, Molecules 27 (2022). https://doi.org/10.3390/molecules27185797.
7) Did the color of compounds affect the bioassays? and how the authors ensure that, specifically, these compounds are colored yellow and the conducted experiments are colorimetric ones.
Response: The color of the compounds did not interfere in the assay’s development, since in the colorimetric analysis as the MTT, for example, the treatments were removed after the period of incubation prior the assay performance. The compounds are not yellow. They are colorless and were reconstituted in cellular medium to keep the consistence in all experiments.
General references:
Note: these references refer to the manuscript, but here they are numbered differently.
[1] T. NAKAYAMA, Y. ENOKI, K. HASHIMOTO, Hydrogen Peroxide Formation during Catechin Oxidation Is Inhibited by Superoxide Dismutase., Food Sci. Technol. Int. Tokyo 1 (1995) 65–69. https://doi.org/10.3136/fsti9596t9798.1.65.
[2] M. Mochizuki, S.I. Yamazaki, K. Kano, T. Ikeda, Kinetic analysis and mechanistic aspects of autoxidation of catechins, Biochim. Biophys. Acta - Gen. Subj. 1569 (2002) 35–44. https://doi.org/10.1016/S0304-4165(01)00230-6.
[3] A.A. Caro, A. Davis, S. Fobare, N. Horan, C. Ryan, C. Schwab, Antioxidant and pro-oxidant mechanisms of (+) catechin in microsomal CYP2E1-dependent oxidative stress, Toxicol. Vitr. 54 (2019) 1–9. https://doi.org/10.1016/j.tiv.2018.09.001.
[4] C. Manosalva, J. Quiroga, A.I. Hidalgo, P. Alarcón, N. Anseoleaga, M.A. Hidalgo, R.A. Burgos, Role of Lactate in Inflammatory Processes: Friend or Foe, Front. Immunol. 12 (2022) 1–14. https://doi.org/10.3389/fimmu.2021.808799.
[5] M. Certo, C.H. Tsai, V. Pucino, P.C. Ho, C. Mauro, Lactate modulation of immune responses in inflammatory versus tumour microenvironments, Nat. Rev. Immunol. 21 (2021) 151–161. https://doi.org/10.1038/s41577-020-0406-2.
[6] Y. Fang, Z. Li, L. Yang, W. Li, Y. Wang, Z. Kong, J. Miao, Y. Chen, Emerging roles of lactate in acute and chronic inflammation, Cell Commun. Signal. 8 (2024) 1–22. https://doi.org/10.1186/s12964-024-01624-8.
[7] L.B. Ivashkiv, The hypoxia–lactate axis tempers inflammation, Nat. Rev. Immunol. 20 (2020) 85–86. https://doi.org/10.1038/s41577-019-0259-8.
[8] K. Yang, J. Xu, M. Fan, F. Tu, X. Wang, T. Ha, D.L. Williams, C. Li, Lactate Suppresses Macrophage Pro-Inflammatory Response to LPS Stimulation by Inhibition of YAP and NF-κB Activation via GPR81-Mediated Signaling, Front. Immunol. 11 (2020) 1–13. https://doi.org/10.3389/fimmu.2020.587913.
[9] H.C. Zhou, W.W. Yu, X.Y. Yan, X.Q. Liang, X.F. Ma, J.P. Long, X.Y. Du, H.Y. Mao, H. Bin Liu, Lactate-driven macrophage polarization in the inflammatory microenvironment alleviates intestinal inflammation, Front. Immunol. 13 (2022) 1–12. https://doi.org/10.3389/fimmu.2022.1013686.
[10] S. Hajjar, X. Zhou, pH sensing at the intersection of tissue homeostasis and inflammation, Trends Immunol. 44 (2023) 807–825. https://doi.org/10.1016/j.it.2023.08.008.
[11] A. Riemann, A. Ihling, J. Thomas, B. Schneider, O. Thews, M. Gekle, Acidic environment activates inflammatory programs in fibroblasts via a cAMP-MAPK pathway, Biochim. Biophys. Acta - Mol. Cell Res. 1853 (2015) 299–307. https://doi.org/10.1016/j.bbamcr.2014.11.022.
[12] K. Rajamäki, T. Nordström, K. Nurmi, K.E.O. Åkerman, P.T. Kovanen, K. Öörni, K.K. Eklund, Extracellular acidosis is a novel danger signal alerting innate immunity via the NLRP3 inflammasome, J. Biol. Chem. 288 (2013) 13410–13419. https://doi.org/10.1074/jbc.M112.426254.
Round 2
Reviewer 1 Report
Comments and Suggestions for Authors
I have no further comments on the content of the manuscript. But I just recommend that the authors once carefully review the manuscript to ensure proper and consistent use of abbreviations. In some sections, I noticed that certain abbreviations were defined more than once. Also, the correct spacing between words in certain terms, such as in "ΔG MM/GBSA."
Comments on the Quality of English LanguageMinor English editing is needed.
Author Response
To: Reviewer
Dear Reviewer,
Thank you for considering our manuscript entitled "Euterpe oleracea Mart. Bioactive Molecules: promising agents to modulate the NLRP3 inflammasome" for publication in Biology (Section: Biology and Function of Inflammasomes). I am sending to you the revised version of our manuscript.
All the reviewers’ comments were carefully considered, and the appropriate changes were applied. We highlighted the changes in red.
We would greatly appreciate an opportunity to publish our manuscript in Biology. Thank you for your time and consideration. Please do not hesitate to let us know if you have any concerns or questions.
Yours Sincerely,
Alencar Kolinski Machado
Ana Cristina Andreazza
Corresponding authors
Reviewer comments:
Comments and Suggestions for Authors: I have no further comments on the content of the manuscript. But I just recommend that the authors once carefully review the manuscript to ensure proper and consistent use of abbreviations. In some sections, I noticed that certain abbreviations were defined more than once. Also, the correct spacing between words in certain terms, such as in "ΔG MM/GBSA."
Response: Thank you for pointing this out. We have carefully revised the manuscript again and all the abbreviations and space between words and terms were checked and, if necessary, corrected.
Comments on the Quality of English Language: Minor English editing is needed.
Response: We have revised the entire manuscript in terms of English quality. All the typos or issues were addressed.